# Ubiquitin ligase STUB1 destabilizes IFNγ-receptor complex to suppress tumor IFNγ signaling

Georgi Apriamashvili[1,6], David W. Vredevoogd [1,6], Oscar Krijgsman[1], Onno B. Bleijerveld [2], Maarten A. Ligtenberg[1], Beaunelle de Bruijn[1], Julia Boshuizen[1], Joleen J. H. Traets[1], Daniela D'Empaire Altimari[1], Alex van Vliet[1], Chun-Pu Lin [1], Nils L. Visser[1], James D. Londino [3], Rebekah Sanchez-Hodge[4], Leah E. Oswalt [4], Selin Altinok [4], Jonathan C. Schisler [4], Maarten Altelaar[2,5] & Daniel S. Peeper [1✉]

The cytokine IFNγ differentially impacts on tumors upon immune checkpoint blockade (ICB). Despite our understanding of downstream signaling events, less is known about regulation of its receptor (IFNγ-R1). With an unbiased genome-wide CRISPR/Cas9 screen for critical regulators of IFNγ-R1 cell surface abundance, we identify STUB1 as an E3 ubiquitin ligase for IFNγ-R1 in complex with its signal-relaying kinase JAK1. STUB1 mediates ubiquitination-dependent proteasomal degradation of IFNγ-R1/JAK1 complex through IFNγ-R1[K285] and JAK1[K249]. Conversely, STUB1 inactivation amplifies IFNγ signaling, sensitizing tumor cells to cytotoxic T cells in vitro. This is corroborated by an anticorrelation between *STUB1* expression and IFNγ response in ICB-treated patients. Consistent with the context-dependent effects of IFNγ in vivo, anti-PD-1 response is increased in heterogenous tumors comprising both wildtype and STUB1-deficient cells, but not full STUB1 knockout tumors. These results uncover STUB1 as a critical regulator of IFNγ-R1, and highlight the context-dependency of STUB1-regulated IFNγ signaling for ICB outcome.

[1] Division of Molecular Oncology and Immunology, Oncode Institute, The Netherlands Cancer Institute, Plesmanlaan 121, 1066 CX Amsterdam, The Netherlands. [2] Proteomics Core Facility, The Netherlands Cancer Institute, Plesmanlaan 121, 1066 CX Amsterdam, The Netherlands. [3] Division of Pulmonary, Critical Care and Sleep Medicine, The Ohio State University Wexner Medical Center, 410 W 10th Avenue, Columbus, OH, USA. [4] McAllister Heart Institute and Department of Pharmacology, The University of North Carolina at Chapel Hill, 111 Mason Farm Rd., 3340 C MBRB CB #7126, Chapel Hill, NC, USA. [5] Biomolecular Mass Spectrometry and Proteomics, Bijvoet Center for Biomolecular Research and Utrecht Institute for Pharmaceutical Sciences, University of Utrecht, and Netherlands Proteomics Center, Padualaan 8, 3584 CH Utrecht, The Netherlands. [6] These authors contributed equally: Georgi Apriamashvili, David W. Vredevoogd. ✉email: d.peeper@nki.nl

Although immune checkpoint blockade (ICB) has been a major clinical success in the treatment of a variety of cancer indications, the majority of patients fail to show durable clinical responses[1,2]. This is caused by both upfront and acquired resistance mechanisms[3–7], for which predictive biomarkers are being actively sought[8–17]. A common resistance mechanism relates to the intrinsic insensitivity that tumors develop against cytokines secreted by cytotoxic T cells, including IFNγ and TNF[4,5,18–20]. IFNγ can promote anti-tumor activity indirectly, by inducing secretion of lymphocyte-attracting chemokines such as CXCL9, CXCL10 and CXCL11, and by skewing the attracted immune infiltrate to be more inflammatory[21–23]. IFNγ can inhibit tumorigenesis also directly, by improving antigen processing and presentation, and by inducing the expression of cell cycle inhibitors, such as p21$^{Cip1}$, and pro-apoptotic proteins, such as caspase 1 and caspase 8[24,25]. Moreover, IFNγ can sensitize tumor cells to other T cell-derived effector cytokines by, for example, increasing the expression of Fas and TRAIL receptors[26,27].

In line with these biological functions, enhanced expression of IFNγ response genes in tumors is associated with better responses to immunotherapy[17,28]. These clinical findings are underscored by preclinical research showing a critical role for IFNγ in hindering tumorigenesis and maintaining tumor control[29]. Conversely, aberrations in the IFNγ response pathway, such as inactivation of JAK1, are associated with resistance to immunotherapy[4,5,18]. Additionally, multiple experimental and clinical approaches have shown that tumor cells benefit from either loss or reduction in IFNγ-receptor (IFNγ-R) levels in the context of ICB therapy[5] or T cell pressure[6,30,31]. But the picture is more complex: recent studies have suggested that IFNγ-insensitive tumors are, counterintuitively, more sensitive to immune-pressure[32]. When admixed with tumor cells proficient in IFNγ signaling however, IFNγ-resistant cells can grow out[32]. Furthermore, prolonged IFNγ exposure can give rise to multifactorial resistance mechanisms and impact the tumor microenvironment (TME)[33,34].

Although the IFNγ signaling pathway has been studied extensively, and different regulatory mechanisms have been uncovered, less is known about cell-intrinsic regulation of IFNγ-R1, its essential ligand-binding receptor chain residing at the tumor cell surface[35,36]. In this study, we therefore perform a genome-wide CRISPR/Cas9 knockout screen to uncover critical factors regulating the abundance of IFNγ-R1 on tumor cells. We focus on top hit STUB1 and mechanistically characterize how it governs IFNγ signaling by regulating its receptor. Lastly, we assess the context-dependent impact of STUB1-dependent IFNγ signaling for the response to anti-PD-1 treatment in vivo.

## Results

### High IFNγ-R1 expression results in increased sensitivity of tumor cells to T cell killing.
Whereas it is established that defects in the IFNγ receptor complex ablate IFNγ tumor signaling[5,29], we hypothesized that, in turn, elevating IFNγ receptor levels may enhance IFNγ signaling. To investigate this, we deployed three complementary approaches. First, using publicly available single cell RNA sequencing data from patient tumors[37–39], we asked whether tumor-intrinsic expression of the IFNγ-R complex components (comprising IFNγ-R1, IFNγ-R2, JAK1, JAK2 and STAT1), whether as a whole complex or each component independently, correlates with an IFNγ response signature in malignant cells (Fig. 1a and Supplementary Fig. 1a). This analysis revealed that the expression of both the complex as a whole and each of the single components correlates with a tumor-intrinsic IFNγ response (Fig. 1a and Supplementary Fig. 1a). Extending

this observation, we also found a positive correlation between IFNγ-R complex expression and IFNγ response signature expression upon treatment with cytotoxic CD8$^+$ T cells in established melanoma cell lines[19] (Fig. 1b).

Third, we investigated whether the increased expression of components of the IFNγ-R complex is causal in establishing a stronger IFNγ response. Because IFNγ-R1 is the essential ligand-binding receptor chain for IFNγ[35,36], we took advantage of the heterogeneity we observed for its expression levels in the human melanoma cell line D10 and FACsorted tumor cells with high and low expression levels of IFNγ-R1 (Fig. 1c, d). As control proteins, we determined the expression of other cell-surface proteins, namely PD-L1 and MHC class I, which were expressed identically in the IFNγ-R1$^{High}$ and IFNγ-R1$^{Low}$ cell populations (Fig. 1e and Supplementary Fig. 1b). We then investigated whether IFNγ-R1$^{High}$ and IFNγ-R1$^{Low}$ cells differentially respond to IFNγ. By flow cytometry, we observed that IFNγ-R1$^{High}$ cells induced PD-L1 to a greater extent upon IFNγ treatment than IFNγ-R1$^{Low}$ cells did. This result indicates that the expression levels of the endogenous IFNγ-R1 protein dictate the strength of the response to IFNγ (Fig. 1e). This effect had also a biological consequence: in a competition experiment, IFNγ treatment was more detrimental to IFNγ-R1$^{High}$ than to IFNγ-R1$^{Low}$ cells (Supplementary Fig. 1c, d).

We repeated this experiment with cytotoxic T cells, employing the matched tumor HLA-A*02:01$^+$/MART1$^+$ and 1D3 TCR T cell system we previously developed[19]. In brief, D10 melanoma cells endogenously express the MLANA-derived antigen MART-1, which they present on HLA-A*02:01. In turn, this enables the cells to be recognized by CD8$^+$ T cells that had been transduced to express the MART-1-specific 1D3 TCR. In this experiment also, D10 IFNγ-R1$^{High}$ melanoma cells showed higher IFNγ-dependent susceptibility to T cell killing than IFNγ-R1$^{Low}$ cells (Fig. 1f, and Supplementary Fig. 1e). Thus, the heterogeneous expression level of IFNγ-R1, even in an established tumor cell line, has a biological consequence, in that higher IFNγ-R1 expression results in increased sensitivity of tumor cells to T cell killing.

### Whole genome CRISPR/Cas9 screen identifies regulators of IFNγ-R1 expression.
For potential future therapeutic exploitation of this observation, we deemed it important to start dissecting the mechanism governing IFNγ-R1 expression in an unbiased fashion. To identify novel regulators of cell-surface-expressed IFNγ-R1, we performed a CRISPR/Cas9 knockout screen (Fig. 1g). Cas9-expressing human D10 melanoma cells were lentivirally transduced with a genome-wide knockout library[40], in duplicate. After 2 days of puromycin selection, we harvested a library reference sample. After an additional 15 days of culturing, we FACsorted both the lower (IFNγ-R1$^{Low}$) and upper (IFNγ-R1$^{High}$) 10% of IFNγ-R1-expressing cell populations (as well as an unsorted bulk reference sample, Fig. 1g). Genomic DNA was isolated and sgRNA sequences were amplified by PCR. Analysis of the DNA sequencing data revealed a strong correlation between biological replicates (Supplementary Fig. 1f). By comparing the library reference with unsorted control samples, we confirmed significant depletion of known essential genes[41] (Supplementary Fig. 1g). These quality control measures illustrate the robustness of the screen.

By MAGeCK analysis[42], we identified several hits affecting IFNγ-R1 expression (Fig. 1h). Comparative analysis of the IFNγ-R1$^{High}$ and IFNγ-R1$^{Low}$ melanoma populations revealed that cells carrying sgRNAs targeting *IFNGR1* were most abundant in the latter population, again confirming the robustness of the screen (Fig. 1h). More interestingly, the E3 ubiquitin ligase STIP1 homology and U-box containing protein 1 (STUB1, also known as CHIP and encoded by *STUB1*) emerged as the strongest hit

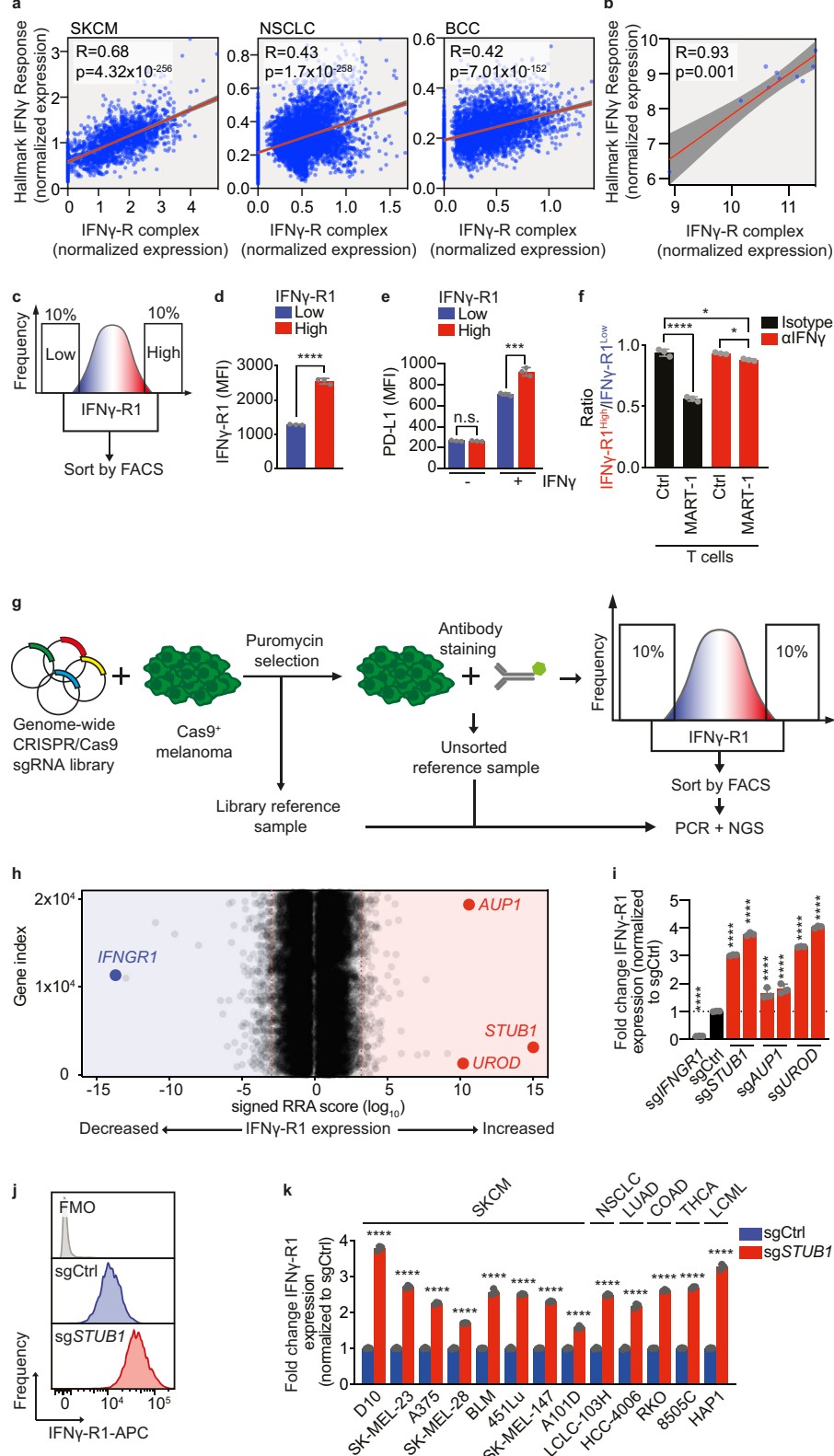

suppressing IFNγ-R1 cell-surface abundance. We also identified other genes negatively affecting IFNγ-R1 expression, including Ancient ubiquitous protein 1 and Uroporphyrinogen Decarboxylase (encoded by *AUP1* and *UROD*, respectively). We performed the same IFNγ-R1 regulator screen in a second human melanoma cell line, SK-MEL-23, which was similar in quality (Supplementary

Fig. 1g) and again identified STUB1 and UROD (Supplementary Fig. 1h).

To validate these screen hits, we inactivated either *STUB1*, *AUP1* or *UROD* using two independent sgRNAs for each gene. Whereas cells expressing sg*IFNGR1* showed a near-complete loss of IFNγ-R1 expression, inactivation of either *STUB1* or *UROD*,

**Fig. 1 Genome-wide CRISPR/Cas9 knockout screen identifies negative regulators of IFNγ-R1 expression to modulate its cell-surface abundance.**
**a** Spearman correlation of IFNγ-R complex expression with Hallmark IFNγ response signature in scRNA sequencing data[37–39]. SKCM skin cutaneous melanoma, $n = 1881$; NSCLC non-small-cell lung cancer, $n = 5716$; BCC, basal cell carcinoma, $n = 3551$. **b** Spearman correlation of IFNγ-R complex expression with Hallmark IFNγ response signature in melanoma cell lines treated with MART-1 T cells[19], $n = 10$. **c** Schematic outline of the FACsorting strategy to establish IFNγ-R1^High and IFNγ-R1^Low D10 human melanoma cell populations. **d** Mean Fluorescence Intensity (MFI) of IFNγ-R1 expression on D10 melanoma cells 2 days after sorting the cells as indicated in **c**. **e** IFNγ-induced PD-L1 expression of IFNγ-R1^High and IFNγ-R1^Low cell populations 24 h after IFNγ (10 ng/ml) treatment. **f** Quantification of the ratio IFNγ-R1^High : IFNγ-R1^Low in competition assay of (Supplementary Fig. 1e). **g** Schematic outline of the FACsort-based genome-wide CRISPR-KO screen to identify genes regulating IFNγ-R1 cell-surface expression. **h** Screen results; red dotted lines indicate FDR cutoff <0.25 for genes enriched in 10% of cells with highest (right) or lowest (left) IFNγ-R1 expression (MAGeCK analysis). Gene names indicate top enriched sgRNAs in cells with the 10% highest IFNγ-R1 expression (right), and sgRNAs targeting *IFNGR1* (left), serving as a positive control. **i** Quantification of IFNγ-R1 expression by flow cytometry on cells expressing the indicated sgRNAs. **j** IFNγ-R1 expression on D10 melanoma cells measured by flow cytometry in cells expressing indicated sgRNAs. FMO fluorescence minus one, APC Allophycocyanin. **k** IFNγ-R1 expression (normalized to each respective sgCtrl) measured by flow cytometry in indicated human tumor cell lines expressing either sgCtrl or sg*STUB1*. SKCM skin cutaneous melanoma, NSCLC non-small-cell lung cancer, LUAD lung adenocarcinoma, COAD colon adenocarcinoma, THCA thyroid carcinoma, LCML chronic myelogenous leukemia. Mean ± SD in **d**, **e**, unpaired *t*-test for three biological replicates. ****$p < 0.0001$ (**d**), ***$p = 0.000467$ (**e**). Mean ± SD in **f**: ****$p < 0.0001$, ordinary one-way ANOVA for three biological replicates, with Tukey's post hoc testing. Mean ± SD in **i**: ****$p < 0.0001$, ordinary one-way ANOVA for three biological replicates with Dunnett post hoc testing. Mean±SD in **k**, ****$p < 0.0001$, multiple *t*-tests for three biological replicates.

and to a lesser extent *AUP1*, instead resulted in a robust increase of IFNγ-R1 abundance (Fig. 1i).

To determine whether STUB1 functions as a negative regulator of IFNγ-R1 expression beyond melanoma, we depleted it by Cas9-mediated knockout from cell lines originating from different tumor indications, and assessed the effect on the expression of IFNγ-R1. We again observed strong induction of IFNγ-R1 expression in all cell lines tested, indicating that STUB1 has a key role in limiting IFNγ-R1 expression across different tumor types (Fig. 1j, k). This conserved regulatory role of STUB1 was underscored by the observation that primary liver and heart tissues from *Stub1*-deficient mice[43] also showed elevated IFNγ-R1 levels (Supplementary Fig. 1i, j).

**STUB1 specifically regulates the cell-surface fraction of IFNγ-R1.** This broad effect prompted us to mechanistically dissect how STUB1 regulates IFNγ-R1 expression. qPCR analysis for *IFNGR1* showed that its transcript levels were indistinguishable between WT and *STUB1*-deficient cells (Supplementary Fig. 2a). Therefore, we focused our attention on a post-transcriptional mode of regulation. We first determined in which cellular compartment STUB1 regulates IFNγ-R1 expression. Cell lysates of *STUB1*-proficient and *STUB1*-deficient cells were treated with various deglycosylating enzymes and analyzed by SDS-PAGE. IFNγ-R1 manifested as multiple, distinguishable protein species. The strongest increase in IFNγ-R1 upon STUB1 depletion was seen in the high molecular weight, EndoH-resistant species. This suggests that the regulation of IFNγ-R1 by STUB1 occurs after the receptor passes through the endoplasmic reticulum (Supplementary Fig. 2b, c).

To determine which of the IFNγ-R1 protein species are located at the tumor cell surface, we performed biotin labeling and immunoprecipitation of cell-surface proteins[44]. This analysis showed that only the high molecular weight species of IFNγ-R1 resides at the plasma membrane (Supplementary Fig. 2d). Together, these results imply that STUB1 specifically regulates the cell-surface fraction of IFNγ-R1, which is in accordance with our flow cytometry findings.

**STUB1 destabilizes IFNγ-R1 in JAK1-dependent and JAK1-independent manners.** STUB1, initially identified as a co-chaperone[45], but in fact a bona fide E3 ubiquitin ligase[46,47], affects protein stability by mediating proteasomal degradation of its client proteins[47–49]. Therefore, and in accordance with our observation that STUB1 loss does not affect *IFNGR1* mRNA

levels, we hypothesized that it destabilizes the IFNγ-R1 protein. To test this, we profiled the proteomes of cells expressing either a non-targeting control sgRNA (sgCtrl) or a *STUB1*-targeting sgRNA (sg*STUB1*) by mass spectrometry. This analysis not only confirmed our observation that *STUB1* inactivation increases IFNγ-R1 levels, but it also revealed a marked increase in the abundance of the JAK1 protein (Fig. 2a). This finding was confirmed in a second cell line (Supplementary Fig. 2e) and validated by immunoblotting for IFNγ-R1 and JAK1 in both cell lines (Fig. 2b, Supplementary Fig. 2f, g). Of note, STUB1 ablation caused only a minor subset of the proteome to be differentially regulated (Fig. 2a and Supplementary Fig. 2e). In line with its mode of regulation of IFNγ-R1 expression, STUB1 also affected JAK1 protein stability, as *JAK1* transcript levels remained unchanged by *STUB1* inactivation (Supplementary Fig. 2h).

While it is known that the interaction of IFNγ-R1 and JAK1 is essential for the signaling functionality of the IFNγ receptor complex[50,51], a potential role of JAK1 in stabilizing IFNγ-R1 levels, and by extension the IFNγ receptor complex, has not been reported. We first investigated whether heightened JAK1 expression would suffice to drive increased IFNγ-R1 protein stability. Ectopically expressed *JAK1* strongly increased IFNγ-R1 protein abundance (Fig. 2c–e), which translated into increased cell-surface expression (Fig. 2d), even more so than ectopically expressed *IFNGR1* (Fig. 2c–e and Supplementary Fig. 2i, j). This result suggests not only that elevated JAK1 protein levels are sufficient to stabilize IFNγ-R1 protein, but also that *JAK1* expression may be crucial in dictating the amount of IFNγ-R1 present on the cell surface; unexpectedly even more so than *IFNGR1* expression itself.

To determine whether elevated JAK1 levels in *STUB1*-inactivated cells account for the rise in IFNγ-R1 abundance, we inactivated JAK1 in a *STUB1*-deficient background (Fig. 2f, g). This genetic epistasis experiment revealed that *STUB1* inactivation was considerably less effective in enhancing IFNγ-R1 expression in *JAK1* KO cells (Fig. 2f, g). These findings together indicate that *STUB1* deficiency promotes IFNγ-R1 stabilization in a largely JAK1-dependent fashion, with a contribution of JAK1-independent regulation.

**STUB1 requires its TPR domain and E3 ubiquitin ligase activity to reduce IFNγ-R1 and JAK1 expression.** For its role as an E3 ubiquitin ligase, STUB1 relies on several domains (Fig. 2h). At its C-terminus, the UBOX domain represents the catalytic domain, while the N-terminus contains a tetratricopeptide tandem repeat (TPR) domain. This is essential for the interaction of

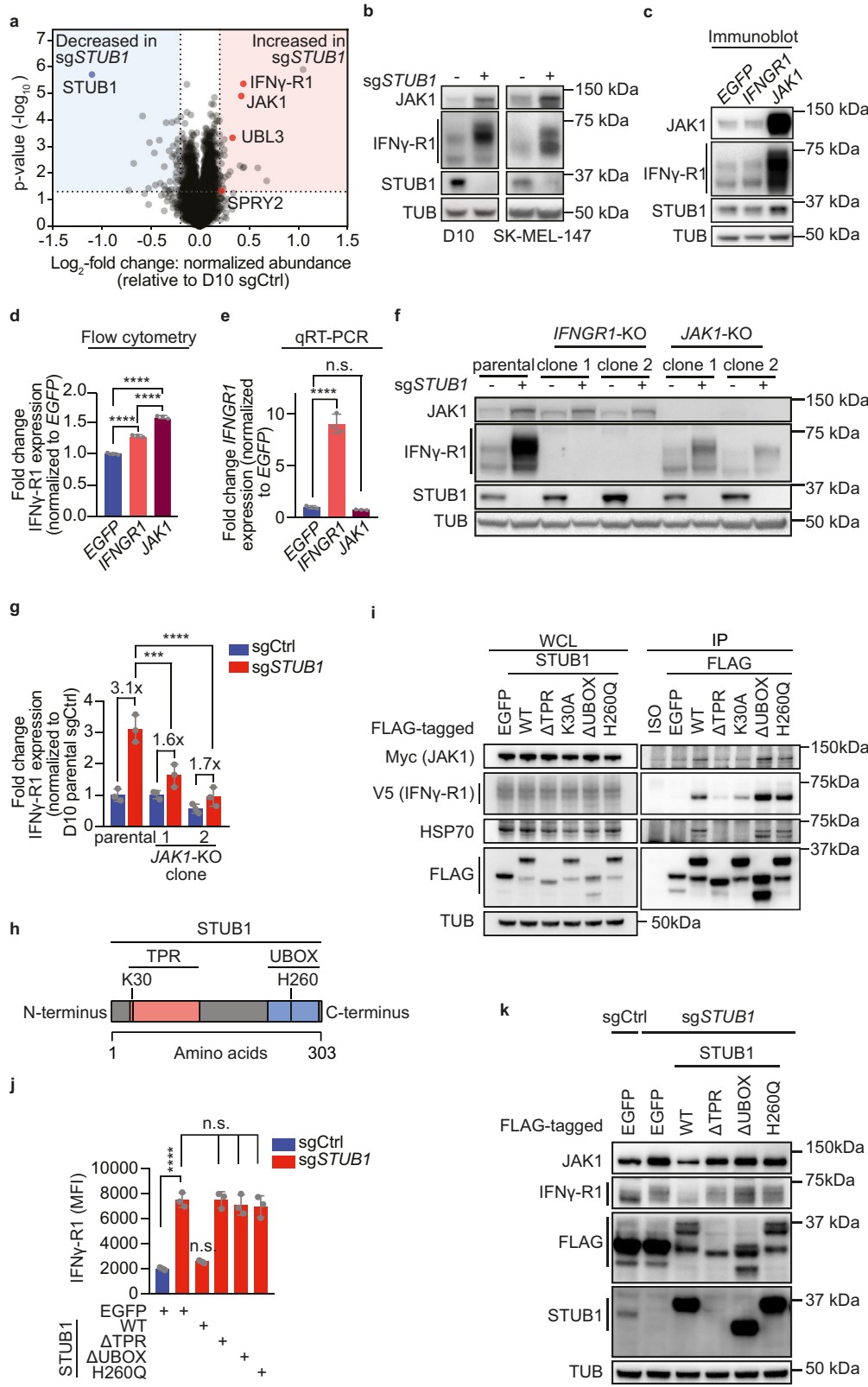

STUB1 with chaperones and, therefore, its substrates[45]. We set out to understand (i) whether STUB1 can interact with IFNγ-R1 and JAK1, (ii) which domains are required for this interaction and (iii) which domains STUB1 requires for regulating the stability of JAK1 and IFNγ-R1.

To address the first question, we co-transfected, into STUB1-deficient HEK293T cells, cDNAs encoding Myc-tagged JAK1, V5-

tagged IFNγ-R1 and either FLAG-tagged full-length STUB1, STUB1 lacking the TPR domain, a STUB1 point mutant abolishing the interaction with chaperones (K30A)[52], STUB1 lacking the UBOX domain or an E3 ligase activity dead mutant of STUB1 (H260Q)[52]. FLAG-EGFP was used as a control. From cell lysates, we pulled down the different STUB1 variants using anti-FLAG antibody and measured co-precipitation of HSP70, as a

**Fig. 2 STUB1 destabilizes cell-surface IFNγ-R1 in JAK1-dependent and JAK1-independent manners. a** Proteomic profiling of D10 cells expressing sgCtrl or sg*STUB1*. Highlighted proteins are differentially regulated in two cell lines (Supplementary Fig. 2e). **b** Immunoblot of D10 (left) and SK-MEL-147 (right) cells lines expressing sgCtrl or sg*STUB1*. Whole cell lysates (WCL) were immunoblotted for indicated proteins (TUB is tubulin). Representative of three biological replicates. **c** Immunoblot of D10 cells ectopically expressing indicated constructs. WCL were immunoblotted for indicated proteins (TUB is tubulin). Representative of three biological replicates. **d** Flow cytometric quantification of IFNγ-R1 expression in D10 cells ectopically expressing indicated constructs. **e** qPCR analysis for *IFNGR1* expression in D10 cells ectopically expressing indicated constructs. *IFNGR1* expression was normalized to *EGFP*-expressing cells using the ΔΔCT method. **f** Immunoblot of parental D10 cells, D10 *IFNGR1*-KO clones and *JAK1*-KO clones expressing sgCtrl or sg*STUB1*. WCL were blotted for indicated proteins (TUB is tubulin). Representative of three biological replicates. **g** Densitometric quantification of IFNγ-R1 protein levels (relative to loading control and normalized to D10 parental sgCtrl-expressing cells) from **f**. **h** Schematic depiction of STUB1 and its functional domains. **i** sg*STUB1*-expressing HEK293T cells were transfected with Myc-tagged JAK1, V5-tagged IFNγ-R1, and the indicated FLAG-tagged STUB1 variants. Left: Immunoblot of the WCL of the indicated samples. Right: Immunoblot of immunoprecipitation (IP) samples. WCL and IP samples were blotted for indicated proteins (TUB is tubulin). ISO Isotype control. **j** Flow cytometric quantification of IFNγ-R1 expression in sgCtrl-expressing D10 cells ectopically expressing EGFP (control) and sg*STUB1*-expressing D10 cells ectopically expressing either EGFP, wildtype STUB1 (WT), or the indicated STUB1 variants. All ectopically expressed proteins were FLAG-tagged. **k** Immunoblot of sgCtrl-expressing D10 cells ectopically expressing EGFP and sg*STUB1*-expressing D10 cells ectopically expressing either EGFP, wildtype STUB1, or the indicated STUB1 variants. WCL were blotted for indicated proteins (TUB is tubulin). Representative of three biological replicates. Mean±SD in **d**, **g**: ordinary one-way ANOVA for three biological replicates with Tukey post hoc testing. ****$p < 0.0001$ (**d**), ***$p = 0.0004$, ****$p < 0.0001$ (**g**). Mean ± SD in (**e**): n.s. $p = 0.8001$, ****$p < 0.0001$, ordinary one-way ANOVA for three biological replicates with Dunnett's post hoc testing. Mean±SD in **j**: ****$p < 0.0001$, ordinary one-way ANOVA for three biological replicates with Sidak's post hoc testing.

positive control for a TPR-dependent STUB1-interacting protein, as well as of IFNγ-R1 and JAK1 (Fig. 2i). We could recapitulate the interaction of STUB1 with HSP70 in TPR domain- and K30-dependent manners (Fig. 2i). The TPR domain as a whole was also required for the interaction with IFNγ-R1 and JAK1, while residue K30 was partially dispensable, albeit required for interacting with JAK1 (Fig. 2i). The UBOX domain and the E3 ubiquitin ligase activity were both dispensable for the interactions (Fig. 2i). These results demonstrate that STUB1 interacts with IFNγ-R1 and JAK1, while suggesting different chaperone requirements for STUB1-IFNγ-R1 and STUB1-JAK1 interactions.

Although the UBOX domain and the E3 ubiquitin ligase activity were not required for the protein interaction, it was important to assess whether they would be required for the STUB1-mediated regulation of IFNγ-R1 and JAK1 protein levels. To test this, we reconstituted either full-length wildtype STUB1, the TPR domain-deficient, UBOX domain-deficient variant or the E3 ligase dead mutant (H260Q) into STUB1-deficient cells (Fig. 2j, k). We observed that only full-length STUB1 was able to reduce IFNγ-R1 and JAK1 protein levels back to wildtype levels (Fig. 2j, k). Together, these results demonstrate that STUB1 requires both its TPR domain and E3 ubiquitin ligase activity to destabilize IFNγ-R1 and JAK1.

**STUB1 drives proteasomal degradation of IFNγ receptor complex through IFNγ-R1$^{K285}$ and JAK1$^{K249}$ residues**. Since STUB1 has been shown to mediate proteasomal degradation of client proteins[48,49], we next asked whether increased protein levels of IFNγ-R1 and JAK1 upon STUB1 inactivation were caused by reduced proteasomal degradation. We treated either wildtype or *STUB1*-deficient cells with MG132, an inhibitor of proteasomal degradation. Western blot analysis of the corresponding cell lysates showed a significant induction of IFNγ-R1 proteins in wildtype cells upon treatment with MG132 (Fig. 3a, b). In contrast, whereas baseline levels of IFNγ-R1 were already increased in *STUB1*-deficient cells, there was no further induction upon MG132 treatment. A similar observation was made for JAK1 (Fig. 3a–c). These results were recapitulated in an additional cell line (Supplementary Fig. 3a–c).

To understand in more detail which lysine residues of IFNγ-R1 and JAK1 are relevant for the STUB1-mediated proteasomal degradation of both factors, we first profiled the ubiquitination levels of lysine (K) residues on JAK1 and IFNγ-R1 in wildtype cells (Supplementary Fig. 3d). We immunopurified peptides

containing a K-ε-diglycine motif; a remnant mark of ubiquitinated proteins after tryptic digestion[53]. The immunoprecipitated peptides were subsequently quantified by mass spectrometry (Supplementary Fig. 3d). From this analysis, we learned that IFNγ-R1$^{K285}$ and JAK1$^{K249}$ are the most frequent targets of ubiquitination.

To determine the relevance of these residues for the STUB1-mediated regulation of IFNγ-R1 and JAK1, we generated melanoma cell clones deficient in both *IFNGR1* and *JAK1* (*IFNGR1*-KO + *JAK1*-KO) in either a wildtype or *STUB1*-deficient background. We then reconstituted *JAK1* and *IFNGR1* either in a wildtype configuration, or in a form in which IFNγ-R1$^{K285}$ and JAK1$^{K249}$ residues were mutated to arginine, thereby precluding ubiquitination events from occurring at those sites (Fig. 3d). We assessed the effects of the various mutations and genotypes on IFNγ-R1 and JAK1 protein levels by flow cytometry and Western blot (Fig. 3e–g and Supplementary Fig. 3e–g). This reconstitution experiment showed that preventing ubiquitination of IFNγ-R1$^{K285}$ and JAK1$^{K249}$ resulted in marked protein stabilization of IFNγ-R1 and JAK1 in wildtype cells (Fig. 3e–g). This increased protein stability of mutant IFNγ-R1$^{K285}$ and JAK1$^{K249}$ occurs through reduced proteasomal turnover, as MG132 treatment was unable to further stabilize IFNγ-R1 and JAK1 levels in the IFNγ-R1$^{K285}$ and JAK1$^{K249}$ mutants, whereas it did in wildtype cells (Fig. 3e–g).

To assess the reliance of STUB1 on these residues for modifying IFNγ-R1 and JAK1 stability, we continued by inactivating STUB1 in the IFNγ-R1$^{K285}$ and JAK1$^{K249}$ mutant cells. We analyzed IFNγ-R1 and JAK1 expression by Western blot (Fig. 3h and Supplementary Fig. 3e, f) and additionally assessed IFNγ-R1 expression by flow cytometry (Fig. 3i and Supplementary Fig. 3g). Whereas in STUB1-proficient cells, the IFNγ-R1$^{K285}$ and JAK1$^{K249}$ mutants resulted in increased stability of IFNγ-R1 and JAK1 (Fig. 3h, i and Supplementary Fig. 3e–g), they were unable to further increase IFNγ-R1 and JAK1 in a *STUB1*-KO background (Fig. 3h, i and Supplementary Fig. 3e–g). This finding suggests that STUB1 requires the lysine residues IFNγ-R1$^{K285}$ and JAK1$^{K249}$ to target their parent proteins, IFNγ-R1 and JAK1, for proteasomal degradation.

To test whether STUB1 can directly ubiquitinate JAK1$^{K249}$, we carried out an in vitro ubiquitination assay, in which STUB1 acts as E3 ubiquitin ligase for a JAK1 fragment (JAK1$^{233-332}$) as substrate. Following the ubiquitination reaction, we analyzed the peptides using mass spectrometry in order to map which residues

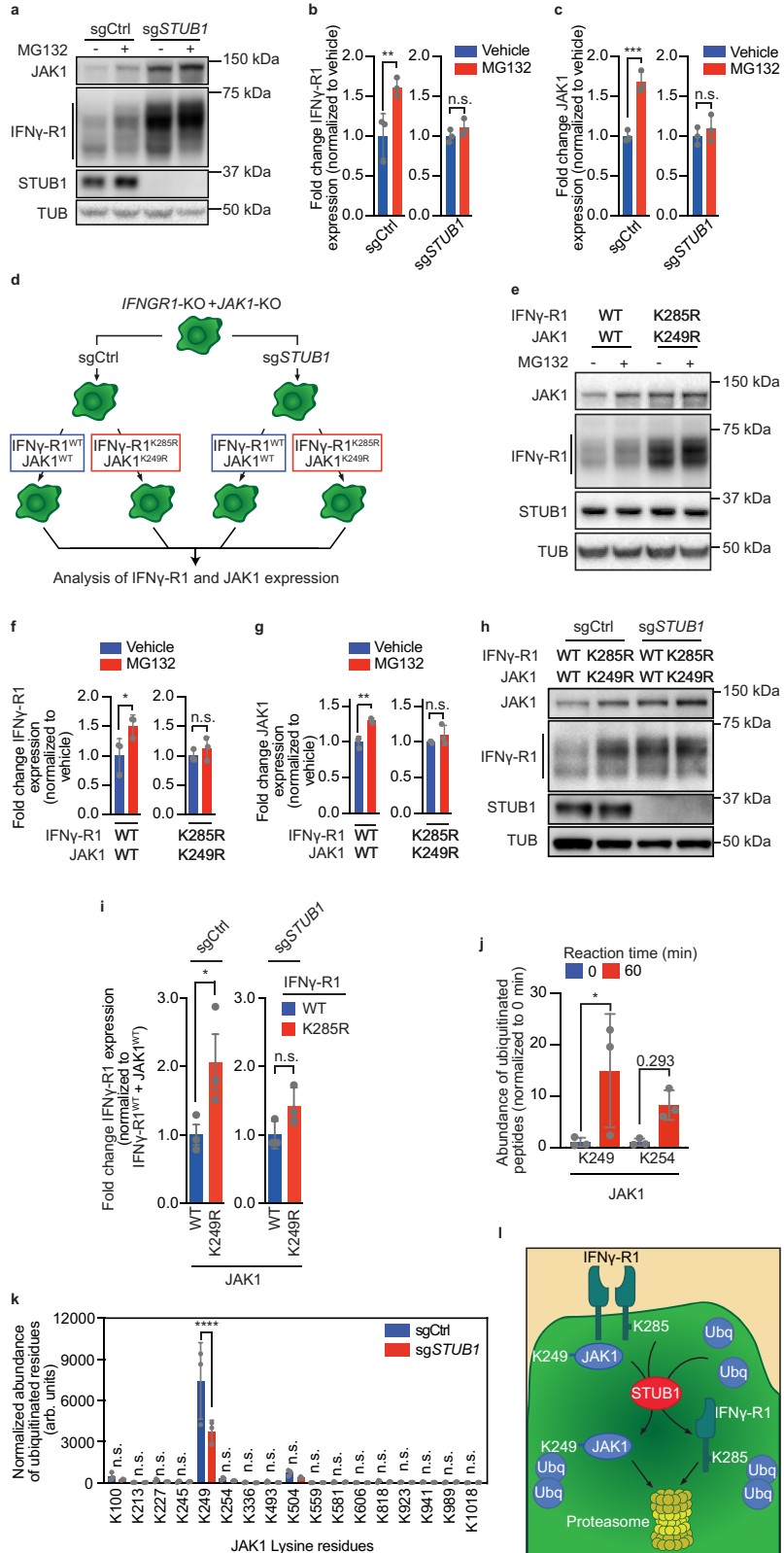

of JAK1$^{233-332}$ were ubiquitinated (Fig. 3j). This analysis revealed that STUB1 is able to directly ubiquitinate JAK1$^{K249}$ and, possibly, JAK1$^{K254}$ (Fig. 3j).

This finding predicts that STUB1-deficient cells exhibit lower ubiquitination levels of JAK1$^{K249}$. To test this hypothesis, we also profiled the ubiquitinated peptides of JAK1 in STUB1-deficient cells. After accounting for the overall increase in JAK1 protein

levels in STUB1-deficient cells, we observed markedly reduced ubiquitination of JAK1$^{K249}$ in cells lacking STUB1 (Fig. 3k). Taken together, these results support a model in which STUB1, through TPR domain-mediated interaction and its E3 ubiquitin ligase activity, regulates proteasomal turnover of IFNγ-R1 and JAK1 protein levels through ubiquitinating K285 and K249 residues, respectively (Fig. 3l).

**Fig. 3 STUB1 drives proteasomal degradation of IFNγ receptor complex through IFNγ-R1$^{K285}$ and JAK1$^{K249}$ residues. a** Immunoblot of D10 cells expressing sgCtrl or sg*STUB1*, treated with vehicle or 10 µM MG132 for 4 h. Whole cell lysates (WCL) were immunoblotted for indicated proteins (TUB is tubulin). Representative of three biological replicates. **b** Densitometric quantification of IFNγ-R1 protein levels (relative to loading control and normalized to vehicle-treated group) from **a**. **c** as in **b** for JAK1 protein. **d** Schematic depiction of reconstitution of IFNγ-R1$^{WT}$ + JAK1$^{WT}$ or IFNγ-R1$^{K285R}$ + JAK1$^{K249R}$ cDNAs in *IFNGR1*-KO + *JAK1*-KO D10 clones in either a sgCtrl- or sg*STUB1*-expressing genetic background. **e** Immunoblot of *IFNGR1*-KO + *JAK1*-KO D10 clones, reconstituted with either IFNγ-R1$^{WT}$ + JAK1$^{WT}$ or IFNγ-R1$^{K285R}$ + JAK1$^{K249R}$ cDNAs, after four-hour treatment with vehicle or 10 µM MG132. WCL were immunoblotted for indicated proteins (TUB is tubulin). Representative of three biological replicates. **f** Densitometric quantification of IFNγ-R1 protein levels (relative to loading control and normalized to vehicle-treated group) from **e**. **g** as in **f** for JAK1 protein. **h** Immunoblot on WCL of *IFNGR1*-KO + *JAK1*-KO D10 clones reconstituted with constructs as outlined in **d**. WCL were immunoblotted for indicated proteins (TUB is tubulin). Representative of three biological replicates. **i** Fold change of IFNγ-R1 MFI (relative to *IFNGR1*-WT + *JAK1*-WT-expressing cells) in *IFNGR1*-KO + *JAK1*-KO D10 clones reconstituted with constructs as outlined in **d**. Bar chart represents an excerpt from Supplementary Fig. 3g. **j** Mass spectrometry-based quantification of STUB1-ubiquitinated JAK1 lysine residues after in vitro ubiquitination reaction of JAK1$^{233-332}$. Depicted lysine residues were also identified in ubiproteome profiling (Supplementary Fig. 3d). **k** Normalized abundance of ubiquitinated JAK1 lysine residues in sgCtrl and sg*STUB1*-expressing D10 cells. **l** Model of STUB1-mediated proteasomal regulation of IFNγ-R1 and JAK1. Mean±SD in **b**, **c**, ordinary one-way ANOVA for three biological replicates with Tukey post hoc testing. **p = 0.0085 (**b**), ***p = 0.0007 (**c**). Mean ± SD in **f**, **g**, ordinary one-way ANOVA for three biological replicates with Sidak post hoc testing. *p = 0.0322 (**f**), **p = 0.0041 (**g**). Mean ± SD in **i**, *p = 0.036, ordinary one-way ANOVA for three biological replicates with Tukey post hoc testing. Mean ±SD in **j**, **k**, ordinary one-way ANOVA for three experimental replicates with Sidak post hoc testing. *p = 0.0346 (**j**), ****p < 0.0001 (**k**).

**STUB1 inactivation sensitizes melanoma cells to cytotoxic T cells through amplified IFNγ signaling.** Having established that STUB1 regulates IFNγ-R1 and JAK1 expression under homeostatic conditions, we next asked whether this regulation affects receptor complex stability during active IFNγ signaling. Whereas wildtype tumor cells moderately upregulated IFNγ-R1 expression upon treatment with increasing amounts of IFNγ, *STUB1*-deficient cells further elevated IFNγ-R1 protein levels, particularly the heavier, cell-surface isoforms (Fig. 4a). We also observed this altered IFNγ response in *STUB1*-deficient cells with downstream mediators of IFNγ signaling, as illustrated by an accelerated and robust onset of STAT1 phosphorylation upon IFNγ treatment in STUB1-depleted cells (Fig. 4b). This altered signaling translated into enhanced transcription of IFNγ-responsive genes, such as *CD274* (encoding PD-L1; Fig. 4c) and *IDO1* (Supplementary Fig. 4a). We confirmed this observation at the protein level (Supplementary Fig. 4b, c).

In light of these results, it was important to assess whether this hyperresponsiveness to IFNγ also alters how STUB1-deficient tumor cells respond to T cell attack. We therefore profiled transcriptomic changes in wildtype and STUB1-deficient melanoma cells 8 h after T cell attack (Supplementary Fig. 4d). Gene set enrichment analysis (GSEA) revealed that STUB1-depleted melanoma cells exhibit an amplified IFNγ response compared to wildtype cells (Fig. 4d and Supplementary Fig. 4e), whereas, as a control for its specificity, genes within the TNF pathway did not show significant enrichment (Fig. 4e). Given these findings, and our previous results demonstrating that elevated IFNγ-R1 levels sensitize tumor cells to IFNγ treatment and cytotoxic T cells, we next tested whether *STUB1* inactivation induces hypersensitivity to (T cell-derived) IFNγ. Indeed, at concentrations where wildtype melanoma cells were barely affected by IFNγ or T cell attack, *STUB1*-deficient melanoma cells were eliminated efficiently (Fig. 4f–i and Supplementary Fig. 4f–i). We confirmed that the sensitization to T cell attack is IFNγ-dependent, as both *STUB1*-deficient and wildtype cells were equally sensitive to T cell attack when lacking IFNγ-R1 expression (Fig. 4j, k, and Supplementary Fig. 4j, k). Collectively, these data show that the strong basal and dynamic induction of IFNγ-R1 expression by *STUB1* inactivation results in intensified IFNγ signaling and consequently, IFNγ-dependent sensitization of melanoma cells to cytotoxic T cells in vitro.

Clinically supporting these findings, we observed a strong negative correlation between *STUB1* expression and the expression of IFNγ response genes in patients undergoing anti-PD-1 treatment (Fig. 4l and Supplementary Fig. 4l). Interestingly, it appears that there are a number of patients with high IFNγ response/low *STUB1* expression who fail to respond to anti-PD-1 blockade.

**STUB1 inactivation enhances IFNγ signaling and increases anti-PD-1 response in heterogeneous tumors with wildtype cells, but not in homogenous STUB1-deficient tumors.** Having observed an enhanced sensitivity of *STUB1*-deficient melanoma cells to cytotoxic T cell-derived IFNγ in vitro (Fig. 4), we next investigated the effects of enhanced IFNγ signaling and its relationship to anti-PD-1 treatment in vivo. We first established *Stub1*-deficient murine melanoma cell lines in which we were able to reiterate our findings from human cell lines in vitro (Supplementary Fig. 5a–e). Importantly and in line with our in vitro data, we validated that immunogenic B16F10-dOVA tumors lacking STUB1 induced PD-L1 to a greater extent than wildtype tumors in vivo (Fig. 5a, b and Supplementary Fig. 5f–h).

To further explore how STUB1 inactivation and the consequentially enhanced IFNγ signaling would impact anti-PD-1 treatment outcome, we employed two relevant preclinical immunotherapy models. First, we used a syngeneic transplantable murine melanoma model, in which we differently labeled wildtype and *Stub1*-deficient B16F10-dOVA cells with either EGFP or mCherry, respectively. We then mixed these cell lines in a 1:1 ratio and transplanted them into immune-deficient NSG mice, or instead into immune-proficient C57BL/6 mice. Animals were treated with either an isotype control antibody or an anti-PD-1 antibody 1 day after tumor inoculation. After 12 days, tumors were harvested and the ratio between wildtype and sg*Stub1* tumor cells was assessed by flow cytometry (Fig. 5c–e). This analysis revealed that while there was a trend towards higher sensitivity of *Stub1*-deficient tumors to immune attack (Fig. 5d, e, compare NSG vs. αISO), strong depletion of *Stub1*-deficient tumors was observed only upon treatment with anti-PD-1 antibody (Fig. 5d, e, compare NSG vs. αPD-1 and αISO vs. αPD-1). This observation is in line with previous reports on the effect of STUB1 inactivation in the context of immunotherapy in a similar mouse tumor model[30] and the effects of differential IFNγ signaling in heterogenous tumors[32].

In contrast, in the second model, in which full *Stub1* knockout B16F10-dOVA tumors were treated once tumors reached 100 mm$^3$, anti-PD-1 responsiveness was not enhanced (Supplementary Fig. 5i). This result is in accordance with the role of STUB1 as a negative regulator of IFNγ signaling and extends previous observations by others on the immune-suppressive effects of

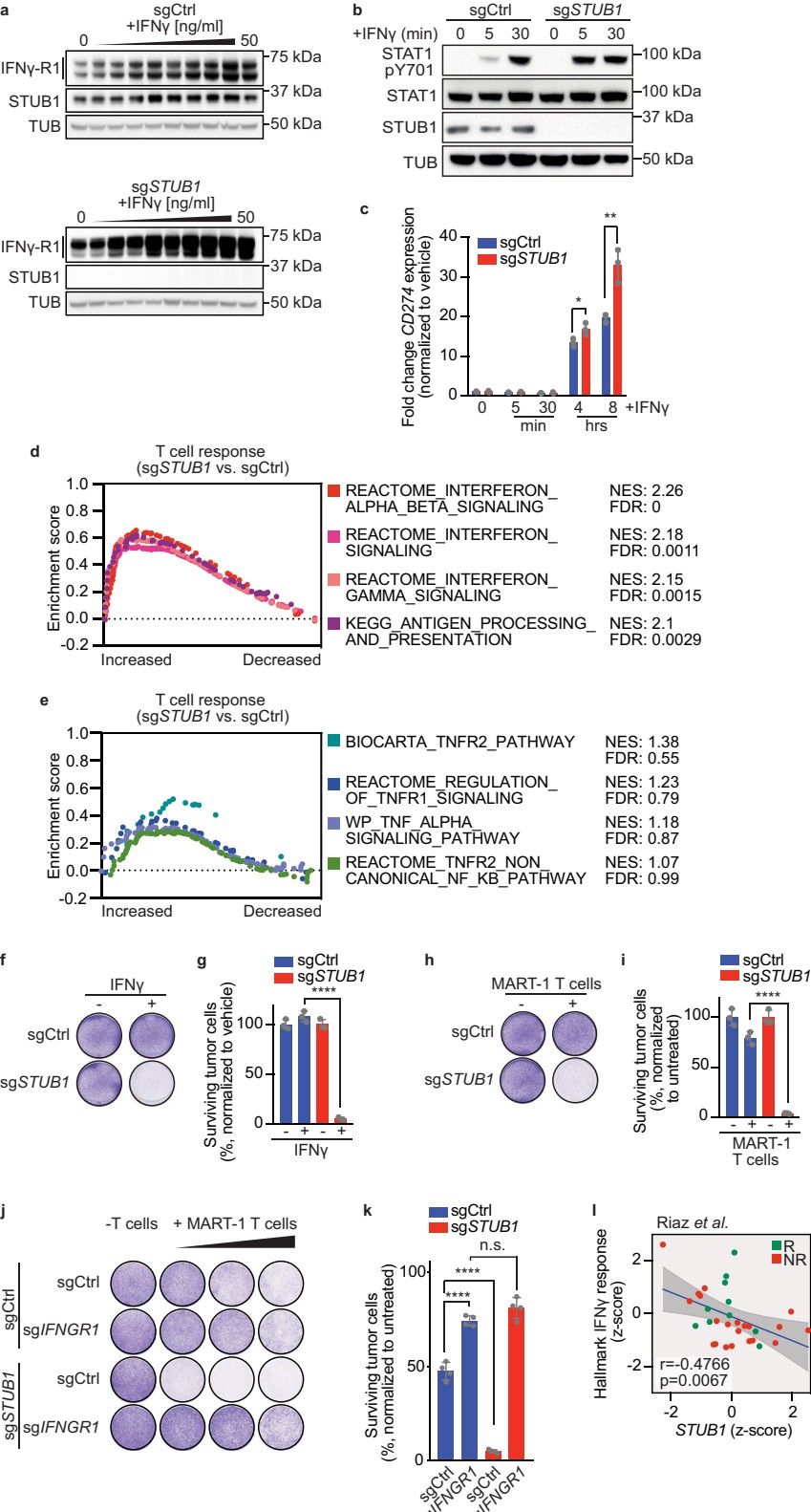

IFNγ signaling on ICB response[32–34]. Taken together, these results on STUB1 increase our understanding of the context-dependent effects of IFNγ signaling, in particular that enhanced IFNγ signaling, through STUB1 inactivation, can improve response to anti-PD-1 in heterogenous tumors, in which also wildtype tumor cells are present, but not homogenous STUB1-deficient tumors.

## Discussion

Although the importance of IFNγ signaling in immunotherapy has become undisputed in recent years, both experimental and preclinical studies have been largely focusing on perturbations in this pathway that contribute to tumor immunogenicity editing and immune escape[4–6,29–31,54]. Considerably less is known about the role and regulation of IFNγ-R1 (cell-surface) expression levels,

**Fig. 4 _STUB1_ inactivation sensitizes melanoma cells to cytotoxic T cells through amplified IFNγ signaling. a** Immunoblots of D10 cells expressing sgCtrl or sg_STUB1_, treated with a two-fold serial dilution of IFNγ (starting at 50 ng/ml) for 30 min. Same protein amounts were loaded on two separate gels, whole cell lysates (WCL) were immunoblotted for indicated proteins (TUB is tubulin) and developed simultaneously. Same exposure for the blots is shown. Representative of three biological replicates. **b** Immunoblot of D10 cells expressing sgCtrl or sg_STUB1_, treated with vehicle or 50 ng/ml IFNγ for the indicated duration. WCL were immunoblotted for the indicated proteins. (TUB is tubulin, pSTAT1 is pY701). Representative of three biological replicates. **c** qPCR analysis of _CD274_ (encoding PD-L1) expression in D10 cells expressing sgCtrl or sg_STUB1_, after treatment with vehicle or 25 ng/ml IFNγ for the indicated duration. **d**, **e** Gene set enrichment analysis on RNA sequencing results for D10 and SK-MEL-147 melanoma cells co-cultured with MART-1 T cells for 8 h (from Supplementary Fig 4d). **d** IFN-related pathways. **e** TNF-related pathways. **f** Colony formation assay of D10 cells expressing sgCtrl or sg_STUB1_ treated with vehicle or 3 ng/ml IFNγ for 5 days. **g** Quantification of colony formation assay in **f**. **h** Colony formation assay of D10 cells expressing sgCtrl or sg_STUB1_ treated with no or MART-1 T cells for 24 h and subsequent culture for 4 days. **i** Quantification of colony formation assay in **h**. **j** Colony formation assay of D10 cells expressing indicated sgRNAs, which were co-cultured with no or MART-1 T cells at T cell: melanoma cell ratios 1:16, 1:8 and 1:4 (left to right) for 24 h and stained 4 days later. **k** Quantification from **j** at a T cell: melanoma cell ratio of 1:8. **l** Spearman correlation of _STUB1_ gene expression with the Hallmark IFNγ response gene set expression in patients tumors[12], n = 31; only anti-PD-1 on-treatment samples were included. Mean ± SD in **c**, **\*\***p = 0.0064, \*p = 0.033, multiple _t_-tests for three biological replicates. Mean ± SD in **g**, **i**, ordinary one-way ANOVA for three biological replicates with Tukey post hoc testing. **\*\*\*\***p < 0.0001. Mean ± SD in **k**, **\*\*\*\***p < 0.0001, n.s. p = 0.1226, ordinary one-way ANOVA for four biological replicates with Tukey post hoc testing.

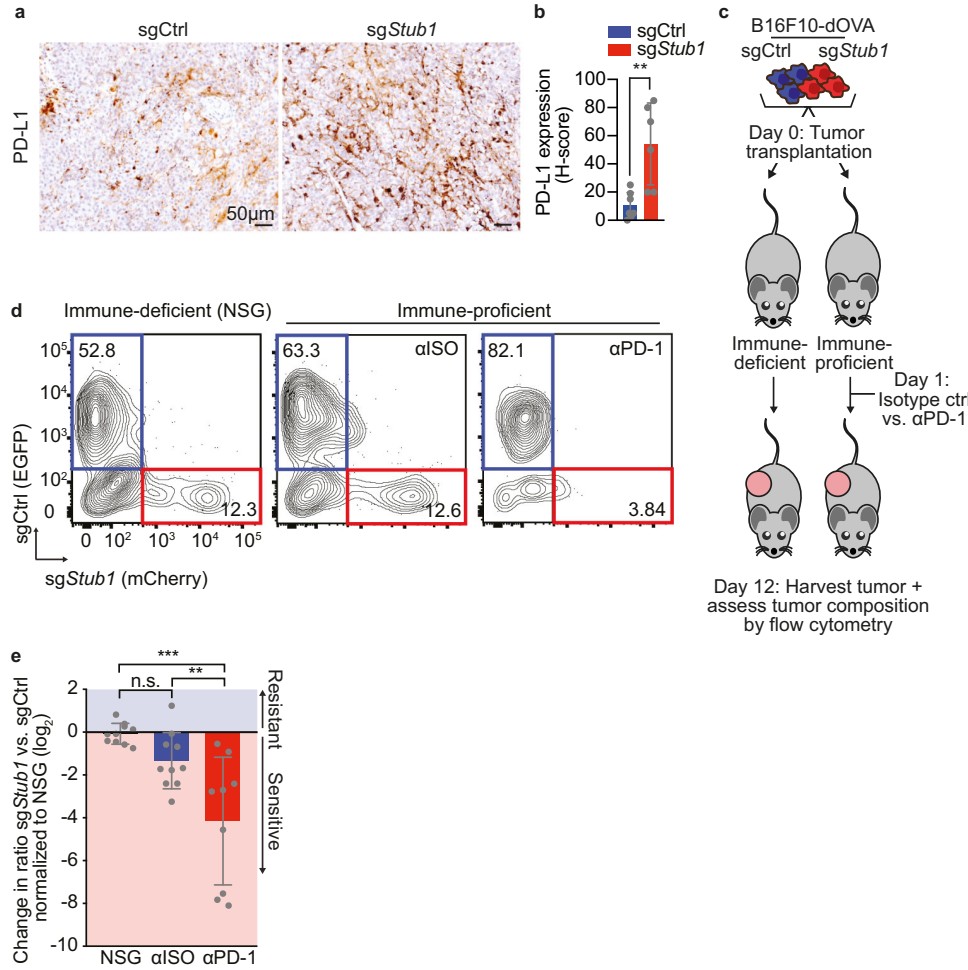

**Fig. 5 STUB1 inactivation enhances IFNγ signaling and increases anti-PD-1 response in heterogeneous tumors with wildtype cells, but not in homogenous STUB1-deficient tumors. a** Immunohistochemistry images of either sgCtrl or sg_Stub1_-expressing B16F10-dOVA tumors in vivo. Tumor samples were stained for PD-L1. **b** Quantification using H-score of PD-L1-positive tumor cells in tumor samples depicted in **a**. **c** Schematic depiction of the in vivo competition assay modeling anti-PD-1 response with B16F10-dOVA cells expressing either sgCtrl or sg_Stub1_, which were differentially labeled with EGFP and mCherry, respectively. **d** Flow cytometry plots from each group of the in vivo experiment outlined in **c** NSG, Isotype control-treated (αISO), anti-PD-1-treated (αPD-1). Number in quadrants indicates % of parent population. **e** Quantification of in vivo competition assay outlined in **c**. Ratios of mCherry vs. EGFP were normalized to the NSG condition. Mean ± SD in **b**, **\*\***p = 0.003, unpaired two-tailed _t_-test, n = 7 for sgCtrl and n = 6 for sg_Stub1_. Mean ± SD in **e**, **\*\*\***p = 0.0002, **\*\***p = 0.0073, n.s. p = 0.2985, ordinary one-way ANOVA with Tukey post hoc testing for n = 10 in NSG and αISO and n = 9 in αPD-1.

for example whether increased abundance sensitizes to (T cell-derived) IFNγ. We show here that in tumor : T cell co-cultures and in patients, increased IFNγ-R complex expression correlates with a stronger IFNγ response. Furthermore, heightening IFNγ-R1 expression levels on tumor cells increases the susceptibility to T cell-derived IFNγ, linking transcriptional IFNγ-dependent signaling in tumors to ICB therapy response[11,17,28].

Given our observations on the effects of differential IFNγ-R1 levels on the strength of IFNγ signaling and its possible impact on anti-PD-1 responses in vivo, we first needed to understand the cell-autonomous regulation of IFNγ-R1 itself. We therefore performed an unbiased genome-wide screen in two cell lines and uncovered STUB1 as the most prominent hit: its loss led to increased IFNγ-receptor complex cell-surface expression. STUB1 acts as an E3 ubiquitin ligase and mediates proteasomal degradation of its core components, IFNγ-R1, and its interaction partner JAK1.

With the identification of the critical STUB1-targeted lysine resides IFNγ-R1$^{K285}$ and JAK1$^{K249}$, we extend previous observations on the ubiquitination of IFNγ-R1[55] and JAK1[56]. The identification of these two residues is of relevance to understand this mode of regulation. IFNγ-R1$^{K285}$ is located in the box 1 motif that is shared among cytokine class II receptors and is critical for JAK1 binding[57]. Conversely, JAK1$^{K249}$ is located in the complementary FERM-domain of JAK1, enabling the binding to the box 1 motif of IFNγ-R1[57]. In combination with our findings of JAK1 being pivotal for IFNγ-R1 stabilization, these observations raise the possibility that JAK1 stabilizes IFNγ-R1 by masking the critical IFNγ-R1$^{K285}$ residue prone to ubiquitination and thereby prevents subsequent STUB1-mediated proteasomal degradation. Interestingly, this ubiquitination-mediated control of IFNγ signaling at the level of IFNγ-R1 may constitute a more common mechanism, since recently another ubiquitin ligase, FBXW7, was implicated in governing IFNγ-R1 signaling in breast cancer[58]. Our findings are complementary to this study: together they not only uncover the importance of ubiquitin-mediated IFNγ-R1 modulation, but also highlight the unexpectedly broad consequences of this type of regulation, with strong effects in tumor cells ranging from heightened immune sensitivity to metastasis. While we demonstrate that STUB1 impacts IFNγ signaling by regulating baseline levels of its receptor complex components, the role of STUB1 for this pathway may be broader, since STUB1 has also been shown to regulate the downstream IFNγ target IRF-1[59].

Our data demonstrate that as a result of IFNγ-R1 stabilization, STUB1 loss leads to enhanced IFNγ response as well as to strong sensitization to cytotoxic T cell-mediated tumor cell killing in vitro. While this sensitizing effect is IFNγ(-R1)-dependent, it does not preclude the cell-intrinsic secondary effects of IFNγ signaling, such as MHC class I upregulation or any of the other anti-tumor effects that have been reported for IFNγ signaling[24–27]. Our data suggest that the physiological role for STUB1 is to dampen the IFNγ response, thereby explaining several previous observations. First, STUB1 inactivation was found to sensitize tumors to immune pressure in the context of GVAX and anti-PD-1 therapy[30]; however, the underlying mechanism of this observation was unknown. Second, in a previous genome-wide loss-of-function screen for IFNγ signaling-independent tumor immune sensitizers, STUB1 was not identified as a hit[19], highlighting its specific role as modulator of IFNγ signaling. Third, STUB1 was identified as a regulator of IFNγ-induced PD-L1 expression[60]. It was postulated that STUB1 directly mediates proteasomal degradation of PD-L1. However, we demonstrate that, instead, STUB1 acts as a modulator of IFNγ signaling and thus indirectly modulates PD-L1 expression.

In clinical trials, PD-1 blockade is being combined with a plethora of secondary treatments[61]. We show that STUB1 loss leads to an enhanced IFNγ-dependent transcriptional program. From a therapeutic point of view this could be beneficial, because several IFNγ target genes, such as HLA, contribute to tumor eradication. However, also PD-L1 represents an established IFNγ target, which we confirm here, and this constitutes an immune-protective tumor trait. Indeed, we show that in homogeneous tumors, STUB1 deficiency failed to improve the response to anti-PD-1, possibly pointing towards IFNγ-mediated immune-suppressive mechanisms[33,34,58,62]. However, in heterogeneous tumors in which wildtype tumor cells were admixed with STUB1-deficient cells, STUB1 deficiency strengthened IFNγ signaling, thereby enhancing the response to anti-PD-1 treatment. These differential effects of STUB1 are in line with the context-dependent effects of IFNγ observed in vivo[62]. For example, Williams et al. demonstrated in similar models that fully IFNγ-insensitive tumors succumb to immune pressure[32]. While more data are needed, there is also clinical evidence suggesting that tumors harboring mutations in the IFNγ signaling pathway can still respond to ICB therapy[63]. Intriguingly, in heterogeneous tumors comprising both IFNγ-sensing and insensitive tumor cells, the latter cells can be protected through bystander PD-L1 expression. This is consistent with the observations described here, showing that STUB1-deficient tumor cells are cleared more effectively by anti-PD-1 treatment when admixed with STUB1-proficient tumor cells.

Collectively, our mechanistic and functional data position STUB1 as a conserved and critical determinant of IFNγ signaling, through its destabilizing effects on both IFNγ-R1 and JAK1. In line with our current understanding of the effects of IFNγ signaling, we demonstrate that heterogeneous STUB1-ablated tumors are relatively more responsive to anti-PD-1 treatment, through blockade of PD-L1-dependent immune evasion mechanisms. Simultaneously, our findings highlight that a more granular understanding of IFNγ signaling (or modulation thereof) will be necessary to fully exploit the anti-tumor effects of STUB1 inhibition, in combination with ICB treatment.

## Methods

**Cell lines used in the study**. The human D10 (female), SK-MEL-23 (female), SK-MEL147 (female), A375 (female), SK-MEL-28 (male), BLM-M (male), 451Lu (male), A101D (male), LCLC-103H (male), HCC-4006 (male), RKO (male), 8505C (female) and HEK293T (female) cell lines were obtained from the internal Peeper laboratory stock, as was the murine B16F10-dOVA (male) cell line. All cell lines were tested monthly by PCR to be negative for mycoplasma infection. Cell lines were authenticated using the STR profiling kit from Promega (B9510).

**MART-1-specific CD8$^+$ T cell generation**. Retrovirus encoding the MART-1-specific T cell receptor was made using a producer cell line as described previously[64]. Peripheral blood was obtained from healthy donors, both male and female, who gave written consent (Sanquin, Amsterdam, the Netherlands). Peripheral blood mononuclear cells (PBMCs) were isolated by density gradient centrifugation using Lymphoprep (Stem cell technologies, #07801). CD8$^+$ T cells were purified from the PBMC fraction using CD8$^+$ Dynabeads (Thermo Fisher Scientific, 11333D) according to manufacturer's instructions. The isolated CD8$^+$ T cells were activated for 48 h on non-tissue culture-treated 24-well-plates, which had been coated with anti-CD3 and anti-CD28 activating antibodies overnight (eBioscience, 16-0037-85, 16-0289-85, each 5 μg per well) at a density of $2 \times 10^6$ cells per well. After 48 h $2 \times 10^6$ cells were harvested and mixed with the MART-1 virus at a 1:1 ratio and plated on a non-tissue culture-treated 24-well-plate, which had been coated with Retronectin overnight (Takara Bio, TB T100B, 25 μg per well). Spinfection was performed for 2 h at $2000 \times g$. 24 h following spinfection, transduced MART-1-specific CD8$^+$ T cells (MART-1 T cells) were harvested and cultured for 7 days, after which the transduction efficiency was assessed by flow cytometry using anti-mouse TCRβ (BD Bioscience, 553174). CD8$^+$ T cells were cultured in RPMI (Gibco, 11879020) containing 10% human serum (One Lamda, A25761), 100 units/ml penicillin, 100 μg per ml Streptomycin, 100 units/ml IL-2 (Proleukin, Novartis), 10 ng/ml IL-7 (ImmunoTools, 11340077) and 10 ng/ml IL-15 (ImmunoTools, 11340157). Following retroviral transduction, cells were maintained in RPMI containing 10% fetal bovine serum (Fisher Scientific, 15605639) and 100 units per ml IL-2.

**In vitro tumor competition assay**. IFNγ-R1^Low and IFNγ-R1^High-expressing tumor cells were labeled with CellTrace CFSE Cell Proliferation Dye (CFSE, Thermo Fisher Scientifc, C34554) or CellTrace Violet Cell Proliferation Dye (CTV, Thermo Fisher Scientific, C34557) according to manufacturer's instructions. The labeled tumor cells were mixed in a 1:1 ratio and $4 \times 10^6$ cells were seeded per 10 cm dish (Greiner). The tumor cell mix was subsequently challenged three times for 24 h with either MART-1 T cells or untransduced control CD8$^+$ T cells at a 1:8 ratio. In parallel, the tumor cell mix was treated with either 25 ng/ml IFNγ or vehicle for 5 days. The surviving tumor cell fraction was analyzed for CFSE and CTV staining by flow cytometry 24 h after the final T cell challenge or after 5 days of IFNγ treatment.

**Antibody dilutions for flow cytometry**. If not stated otherwise, all antibodies for flow cytometry were used at a dilution of 1:100.

**IFNγ-induced PD-L1 and MHC class I expression**. Tumor cells were seeded in 24-well-plates at a density of $3 \times 10^5$ cells per well and treated either with a serial dilution series of IFNγ (PeproTech, 300-02) (starting at 50 ng/ml in two-fold dilution steps) or vehicle for 24 h. The cells were harvested after treatment and stained for PD-L1 (eBioscience, 12-5983-42) and MHC class I (R&D Systems, FAB7098G). Induction of the respective proteins was analyzed by flow cytometry.

**Lentiviral transductions**. HEK293T cells were co-transfected with pLX304 plasmids containing constructs of interest and the packaging plasmids pMD2.G (Addgene, #12259) and psPAX (Addgene, #12260) using polyethylenimine. 24 h after transfection, the medium was replaced with OptiMEM (Thermo Fisher, 31985054) containing 2% fetal bovine serum. Another 24 h later, lentivirus-containing supernatant was collected, filtered and stored at −80 °C. Tumor cells were lentivirally transduced by seeding $5 \times 10^5$ cells per well in a 12-well plate (Greiner), adding lentivirus at a 1:1 ratio. After 24 h the virus-containing medium was removed and transduced tumor cells were selected with antibiotics for at least 7 days.

**Sort-based genome-wide CRISRP/Cas9 knockout screen**. D10 and SK-MEL-23 melanoma cells were first transduced to stably express Cas9 (lentiCas9-Blast, Addgene, #52962) and selected with blasticidin (5 μg/ml) for at least 10 days. The respective cell lines were subsequently transduced with the human genome-wide CRISPR-KO (GeCKO, Addgene, #1000000048, #1000000049) sgRNA library at a 1000-fold representation and a multiplicity of infection of <0.3 to ensure one sgRNA integration per cell. The library transduction was performed in two replicates per cell line. Transduced cells were selected with puromycin (1 μg/ml) for 2 days, after which library reference samples were harvested. Cells were cultured for an additional 15 days to allow gene inactivation and establishment of the respective phenotype. Before sorting, a pre-sort bulk population was harvested. Library-transduced cells were then harvested and stained with anti-IFNγ-R1/CD119-APC antibody (Miltenyi Biotech, 130-099-921) for FACSorting. From the live cell population 10% of cells with the highest and 10% of cells with the lowest IFNγ-R1 expression were sorted. The sorted cells were washed with PBS and the cell pellet was snap frozen. Genomic DNA was isolated using the Blood and Cell culture MAXI Kit (Qiagen, 13362), according to manufacturer's instructions. sgRNAs were amplified using a one-step barcoding PCR using NEBNext High Fidelity 2X PCR Master Mix (NEB, M0541L) and the following primers:

Forward primer:
5'-AATGATACGGCGACCACCGAGATCTACACTCTTTCCCTACACGACG CTCTTCCGATCTNNNNNNNGGCTTTATATATCTTGTGGAAAGGACGAA ACACC-3'

Reverse Primer:
5'-CAAGCAGAAGACGGCATACGAGATCCGACTCGGTGCCACTTTTTCA A-3'

The hexa-N nucleotide stretch contains a unique barcode to identify each sample following deep sequencing. MAGeCK (v0.5.7) was used to perform the analysis of the screen. To assess the depletion of core essential genes we compared the library reference sample to the pre-sorted bulk population. Putative regulators of IFNγ-R1 were identified by comparing the sgRNA abundance among the 10% highest and lowest IFNγ-R1-expressing populations and a signed robust rank aggregation (RRA) score was assigned to the respective genes. sgRNA targets with a false discovery rate (FDR) < 0.25 were considered as putative hits. The MAGeCK input files for the screens in D10 and SK-MEL-23 cells can be found in Supplementary Data 1 and 2, respectively.

**qPCR-based detection of transcriptomic differences**. RNA from D10, SK-MEL-147 and SK-MEL-23 melanoma cells expressing either sgCtrl or sgSTUB1 was isolated using the Isolate II RNA Mini Kit (Bioline, BIO-52072) according to manufacturer's instructions. cDNA was reverse transcribed using the Maxima First Strand cDNA synthesis kit (Fisher Scientific, 15273796) according to manufacturer's instructions. cDNA samples were probed for the expression of *RPL13*, *IFNGR1*, *JAK1*, *CD274* and *IDO1* using the following primers:

*RPL13*:
Forward: 5'-GAGACAGTTCTGCTGAAGAACTGAA-3'

Reverse: 5'-TCCGGACGGGCATGAC-3'
*IFNGR1*:
Forward: 5'-CGGAAGTGACGTAAGGCCG-3'
Reverse: 5'-TTAGTTGGTGTAGGCACTGAGGA-3'
*JAK1*:
Forward: 5'-TACCACGAGGCCGGGAC-3'
Reverse: 5'-AGAAGCGTGTGTCTCAGAAGC-3'
*CD274*:
Forward: 5'-TGGCATTTGCTGAACGCATTT-3'
Reverse: 5'-AGTGCAGCCAGGTCTAATTGTT-3'
*IDO1*:
Forward: 5'-AATCCACGATCATGTGAACCCA-3'
Reverse: 5'-GATAGCTGGGGGTTGCCTTT-3'
Gene Expression was quantified using the SensiFAST SYBR Hi-Rox Kit (Bioline, 92090) in combination with the StepOnePlus Real-Time PCR System (Thermo Fisher). Gene expression was normalized to *RPL13* expression using the ΔΔCt approach.

**T cell-melanoma cell co-culture**. Depending on the melanoma cell line, $5 \times 10^4$ to $1.2 \times 10^5$ cells were seeded per well in 12-well plates in 0.5 ml DMEM containing 10% FBS. Melanoma cells were subsequently either co-cultured with the equivalent amount of control T cells or a serial dilution of MART-1 T cells in 0.5 ml DMEM containing 10% FBS (starting with a 1:1 ratio and two-fold dilution steps). After 24 h T cells were removed by washing the plates with PBS, fresh culture medium was added and the melanoma cells were grown for 4 days. After the control T cell-treated well reached >80% confluence, the medium was removed and all wells were fixed with methanol (50% in $H_2O$) and stained with crystal violet (0.1% in $H_2O$) for 30 min.

B16F10-OVA cells were seeded at a density of $5 \times 10^4$ cells per well in 0.5 ml DMEM containing 10% FBS in 12-well plates. OT-I T cells were then added in a two-fold serial dilution starting from 4:1 (T cell: melanoma cell) ratio in 0.5 ml DMEM containing 10% FBS. After 48 h OT-I T cells were removed by washing the wells with PBS. The remaining melanoma cells were grown for an additional 48 h, before being fixed with methanol (50% in $H_2O$) and stained with crystal violet (0.1% in $H_2O$). The crystal violet was removed and the plates were washed with water. After image acquisition, the crystal violet was suspended using a 10% acetic acid (in $H_2O$) solution and the optical density of the resulting suspension was quantified.

**Protein expression analysis by immunoblot**. Whole cell lysates were generated by removing culture medium and washing the adherent cells on the plate twice with PBS. The cells were then scraped, harvested in 1 ml PBS and pelleted by centrifugation at $1000 \times g$. After removing PBS, the cell pellet was resuspended into the appropriate amount of RIPA lysis buffer (50 mM TRIS pH 8.0, 150 mM NaCl, 1% Nonidet P40, 0.5% sodium deoxycholate, 0.1% SDS) supplemented with HALT Protease and Phosphatase inhibitor cocktail (Fisher Scientific, 78444). Lysis was performed on ice for 30 min. The samples were subsequently centrifuged at $17,000 \times g$ and whole cell lysates were collected. The protein content of each lysate was quantified using Bio-Rad protein assay (Bio-Rad, 500-0006). Protein concentrations were equalized and immunoblot samples were prepared through addition of 4xLDS sample buffer (Fisher Scientific, 15484379) containing 10% β-Mercaptoethanol and subsequent incubation of the samples at 95 °C for 5 min. Proteins in lysates were size-separated using 4–12% Bis-Tris polyacrylamide-SDS gels (Life Technologies) and blotted onto nitrocellulose membranes (GE Healthcare). Blots were blocked using 4% milk powder in 0.2% Tween-20 in PBS. Blocked membranes were incubated with primary antibodies overnight. Immunoblots were developed using Super Signal West Dura Extended Duration Substrate (Thermo Fisher, 34075). Luminescence signal was captured by Amersham Hyperfilm high performance autoradiography film or by the Bio-Rad ChemiDoc imaging system. The following primary antibodies were used anti-IFNγ-R1 (Santa Cruz Biotechnology, sc-28363, dilution: 1:200), anti-JAK1 (D1T6W, Cell Signaling Technology, 50996, dilution: 1:1000), anti-STUB1/CHIP (C3B6, Cell Signaling Technology, 2080, dilution: 1:1000), anti-Tubulin (DM1A, Sigma Aldrich, T9026, dilution: 1:1000), anti-STAT1 (D1K9Y, Cell Signaling Technology, 12994, dilution: 1:1000), anti-STAT1-Tyr701 (58D6, Cell Signaling Technology, 9167, dilution: 1:1000), anti-mouse PD-L1 (MIH5, Thermo Fisher Scientific, 14-5982-81, dilution: 1:1000). IFNγ-R1 is detected as multiple glycosylated forms indicated by a vertical line on the left of each blot.

**Quantification of protein expression of immunoblots**. Protein expression on immunoblots was quantified on 8-bit gray-scale-transformed.tiff images of either scanned Amersham Hyperfilm MP (GE Healthcare, 28906838) or.tiff images obtained by the Bio-Rad ChemiDoc imaging system. Fiji ImageJ was used to select a region of interest for the respective proteins for densitometric analysis. Protein expression for each protein was normalized to the loading control of the respective sample.

**Biotin labeling of cell-surface proteins**. Biotin labeling of cell-surface proteins was performed according to the published protocol by Huang[44]. In brief, $2 \times 10^6$

D10 melanoma cells were seeded in 10 cm culture dish 48 h prior to the experiment. Cells were washed twice in ice-cold PBS/CaCl₂/MgCl₂ (+2.5 mM CaCl₂, 1 mM MgCl₂, pH 7.4). Cell-surface proteins were labeled with 2 ml of 0.5 mg/ml Sulfo-NHS-SS-biotin (in PBS/CaCl₂/MgCl₂) on ice for 30 min. Labeling was quenched by washing cells three times with 3 ml of 50 mM glycine (in PBS/CaCl₂/MgCl₂). Cells were lysed using RIPA lysis buffer and biotinylated proteins were pulled down using Streptavidin-coated magnetic beads. Samples were analyzed as described above.

**Immunoprecipitation.** HEK293T cells were transfected with the indicated cDNAs (*IFNGR1* 5 μg, *JAK1* 20 μg, *STUB1* 5 μg of vector DNA) using polyethylenimine (4.5 μg/μg DNA). Cells were harvested 24 h after transfection, washed in PBS and lysed using NP-40 buffer (1% NP40, 150 mM NaCl, 10 mM TrisHCl, pH = 7.4), supplemented with HALT Protease and Phosphatase inhibitor cocktail (Fisher Scientific, 78444), for 30 min on ice. At least 1 mg of protein was used per pull down. Lysates were incubated with 10 μg of the IP antibodies for 2 h at 4 °C and subsequently pulled down using 120 μl Biorad Surebeads Prot A (1614013) for 2 h at 4 °C. Immunoblots were performed as described above.

**Proteome profiling.** sgCtrl- and sg*STUB1*-expressing D10 and SK-MEL-147 melanoma cells (triplicates for both conditions) were lysed in 8 M urea lysis buffer in the presence of cOmplete Mini protease inhibitor (Roche) and aliquots of 200 μg protein were reduced, alkylated with chloroacetamide, predigested with Lys-C (Wako) (1:75, 4 h at 37 °C) and trypsin-digested overnight (Trypsin Gold, Mass Spectrometry Grade, Promega; 1:50 at 37 °C). Peptide samples were desalted using C18 Sep-Pak cartridges (3cc, Waters) and eluted with acidic 40% and 80% acetonitrile. Dried D10 and SK-MEL-147 digests were reconstituted in 50 mM HEPES buffer and replicates were labeled with 10-Plex TMT reagent (Thermo Fisher Scientific) according to the manufacturer's instructions. Labeled samples were mixed equally for both cell lines, desalted using Sep-Pak C18 cartridges and fractionated by basic reversed-phase (HpH-RP) HPLC separation on a Phenomenex Gemini C18 analytical column (100 × 1 mm, particle size 3 μm, 110 Å pores) coupled to an Agilent 1260 HPLC system over a 60 min gradient. Per cell line, fractions were concatenated to 12 fractions for proteome analysis.

Peptide fractions were analyzed by nanoLC-MS/MS on a Thermo Orbitrap Fusion hybrid mass spectrometer (Q-OT-qIT, Thermo Scientific) equipped with an EASY-NLC 1000 system (Thermo Scientific). Samples were directly loaded onto the analytical column (ReproSil-Pur 120 C18-AQ, 1.9 μm, 75 μm × 500 mm, packed in-house). Solvent A was 0.1% formic acid/water and solvent B was 0.1% formic acid/80% acetonitrile. Samples were eluted from the analytical column at a constant flow of 250 nl/min in a 4-h gradient containing a 120-min increase to 24% solvent B, a 60-min increase to 35% B, a 40-min increase to 45% B, 20-min increase to 60% B and finishing with a 15-min wash. MS settings were as follows: full MS scans (375–2000 m/z) were acquired at 120,000 resolution with an AGC target of 4 × 10⁵ charges and maximum injection time of 50 ms. The mass spectrometer was run in top speed mode with 3 s cycles and only precursors with charge state 2–7 were sampled for MS2 using 60,000 resolution, MS2 isolation window of 1 Th, 5 × 10⁴ AGC target, a maximum injection time of 60 ms, a fixed first mass of 110 m/z and a normalized collision energy of 33%. Raw data files were processed with Proteome Discoverer 2.2 (Thermo Fisher Scientific) using a Sequest HT search against the Swissprot reviewed human database. Results were filtered using a 1% FDR cut-off at the protein and peptide level. TMT fragment ions were quantified using summed abundances with PSM filters requiring a S/N ≥ 10 and an isolation interference cutoff of 35%. Normalized protein and peptide abundances were extracted from PD2.2 and further analyzed using Perseus software (ver. 1.5.6.0)[65]. Differentially expressed proteins were determined using a *t*-test (cutoffs: *p* < 0.05 and LFQ abundance difference < −0.2 ^ > 0.2).

**Ubiquitination site profiling.** For ubiquitination site profiling, D10 melanoma cells expressing either sgCtrl or sg*STUB1* were lysed in 8 M urea lysis buffer in the presence of cOmplete Mini protease inhibitor (Roche). Triplicates corresponding to 14 mg protein per sample for sgCtrl and sg*STUB1*-expressing D10 cells were reduced, alkylated with chloroacetamide, predigested with Lys-C (Wako) (1:75, 4 h at 37 °C) and trypsin digested overnight (Trypsin Gold, Mass Spectrometry Grade, Promega; 1:50 at 37 °C). Peptide samples were desalted using C18 Sep-Pak cartridges (3cc, Waters) and eluted with acidic 40% and 80% acetonitrile. At this stage, aliquots corresponding to 200 μg protein digest were collected for proteome profiling, the remainder of the eluates being reserved for enrichment of ubiquitinated peptides. All peptide fractions were vacuum dried and stored at −80 °C until further processing. Ubiquitinated peptides were enriched by immunoaffinity purification using the PTMScan Ubiquitin Remnant Motif (K-ε-GG) Kit (Cell Signaling Technology, 5562) according to the manufacturer's instructions. Ubiquitinated peptide samples were analyzed by nanoLC-MS/MS on an Orbitrap Fusion Tribrid mass spectrometer equipped with a Proxeon nLC1000 system (Thermo Scientific) using a non-linear 210 min gradient as described previously[66]. Raw data files were processed with MaxQuant (ver. 1.5.6.0)[67], searching against the human reviewed Uniprot database (release 2018_01). False discovery rate was set to 1% for both protein and peptide level and GG(K) was set as additional variable modification for analysis of ubiproteome samples. Ubiquitinated peptides were

quantified with label-free quantitation (LFQ) using default settings. LFQ intensities were Log₂-transformed in Perseus (ver. 1.5.6.0)[65], after which ubiquitination sites were filtered for at least two valid values (out of 3 total) in at least one condition. Missing values were replaced by an imputation-based normal distribution using a width of 0.3 and a downshift of 1.8. Determination of differentially ubiquitinated lysine residues on JAK1 was performed as follows: LFQ values of JAK1 in the global proteome dataset were first normalized to the average LFQ score of housekeeping proteins[68] in wildtype and STUB1-deficient D10 melanoma cells. Similarly, JAK1 peptides identified in the ubiproteome dataset for each genotype were also normalized to the average LFQ scores of housekeeping proteins identified in this dataset. The relative abundance of JAK1 peptides identified in the ubiproteome were subsequently corrected for the normalized abundance of JAK1 in either sgCtrl or sg*STUB1*-expressing cells in the total proteome before plotting the LFQ values.

**In vitro ubiquitination assay.** The in vitro ubiquitination assay was carried out using Human CHIP Ubiquitin Ligase Kit (R&D, K-280) and recombinant human JAK1²³³⁻³³²-GST (N-Term) protein (Novus Biologicals, H00003716-Q01-10 μg). The reaction was carried out according to the manufacturer's instructions and under denaturing conditions. Abundance of ubiquitinated peptides was subsequently measured by mass spectrometry.

**Proteasomal inhibitor treatment.** Melanoma cells were seeded and grown to 80% confluence and treated with either DMSO (vehicle) or with 10 μM MG132 (Medchem Express, HY-13259) for 4 h. The medium was removed 4 h later, cells were washed three times with PBS and whole cell lysates were prepared as described above.

**Animal studies.** All animal studies were approved by the animal ethics committee of the Netherlands Cancer Institute (NKI) and performed in accordance with ethical and procedural guidelines established by the NKI and Dutch legislation. Male mice, of either C57BL/6 (Janvier) or NSG-*B2m*⁻/⁻ (The Jackson Laboratory, 010636; RRID:ISMR_JAX:010636) mouse strains were used at an age of 8–12 weeks. The number of mice per experiment are mentioned in the respective figure legends.

**Animal husbandry.** Mice were housed in IVC cages (Innovive Disposable IVC Rodent Caging System) or isolators, in which HEPA filtered air is provided. IVC cages and the accompanying water bottles and cage enrichment are one-time use and recyclable. Cages arrive securely double-bagged, irradiated, pre-bedded and ready for use. Cages, bedding, cage enrichment and water bottles used in the isolators are autoclaved or irradiated before use. Food is irradiated before use.

Cages were changed once per 1 or 2 weeks. Each rack was handled as a microbiological unit. In between units, surfaces were disinfected and clean materials were used. Light/dark cycle is 12 h.

Once a week, mice were examined for health and welfare issues. Observations were recorded and controlled daily. Animals with health and welfare issues were observed daily. Every day all cages were checked for sufficient water and food. Air humidity (55%), temperature (21 °C) and the light cycle of every room were controlled and recorded.

**In vivo tumor competition assay.** B16F10-dOVA cells were lentivirally transduced with lenti-Cas9-blast to stably express Cas9 and selected with blasticidin (5 μg/ml) for at least 10 days. The cells were then lentivirally transduced to stably express either sgCtrl or sg*Stub1* (lentiGuide-Puro, #52963) and cultured with puromycin (1 μg/ml) for at least 10 days to allow for selection of cells with genetic inactivation of *Stub1*. Knockout efficiency was assessed by immunoblotting. sgCtrl-expressing cells were transduced to stably express EGFP (pLX304-EGFP-Blast) and sg*Stub1*-expressing cells were transduced to stably express mCherry (pLX304-mCherry-Blast). EGFP and mCherry-positive populations were sorted and cultured. Cells were mixed in a 1:1 ratio prior to injection and 5 × 10⁵ cell per mouse were injected into immune-deficient NSG-*B2m*⁻/⁻ (*n* = 10, The Jackson Laboratory, 010636; RRID:ISMR_JAX:010636), or C57BL/6 J mice (*n* = 20, Janvier, C57BL/6 JRj). Tumor-bearing C57BL/6J mice were treated with either 100 μg/mouse isotype control antibody (Leinco Technologies, R1367) or with 100 μg/mouse mouse-PD-1 antibody (Leinco Technologies, P372) 1 and 6 days post tumor injection. Tumors were harvested at day 12 and dissociated into single cell suspensions. Cells were subsequently stained for immune cells using anti-CD45-APC (Miltenyi, 130-102-544) and the tumor composition was analyzed by flow cytometry.

**Anti-PD-1 treatment of B16F10-dOVA tumors.** B16F10-dOVA cells were lentivirally transduced with lenti-Cas9-blast to stably express Cas9 and selected with blasticidin (5 μg/ml) for at least 10 days. The cells were then lentivirally transduced to stably express either sgCtrl or sg*Stub1* (lentiGuide-Puro, #52963) and cultured with puromycin (1 μg/ml) for at least 10 days to allow for selection of cells with genetic inactivation of *Stub1*. Knockout efficiency was assessed by immunoblotting. 5 × 10⁵ cells were injected per mouse on each flank. Once tumors reached an average tumor size of 100 mm³, mice were randomized into the different treatment

groups and subsequently treated with either 100 μg/mouse isotype control antibody (Leinco Technologies, R1367) or with 100 μg/mouse anti-mouse-PD-1 (Leinco Technologies, P372) twice-weekly. Tumor growth was monitored until the tumors reached the humane endpoint (1500 mm$^3$).

**Transcriptomic profiling of melanoma cells after T cell attack**. $2 \times 10^6$ D10 and SK-MEL-147 melanoma cells were plated per dish in 10 cm cell culture dishes 48 h prior to T cell challenge. Melanoma cells were subsequently challenged with either Ctrl or MART-1 T cells for 8 h. The T cells were removed by washing the plates with PBS. The remaining tumor cells were harvested and lysed in RLT buffer (Qiagen, 79216) and sequenced on an Illumina HiSeq2500. Fastq files were mapped to the human reference genome (Homo.sapiens.GRCh38.v77) using Tophat v2.1[69] with default settings for single-end data. The samples were used to generate read count data using itreecount (github.com/NKI-GCF/itreecount). Normalization and statistical analysis of the expression of genes was performed using DESeq2 (V1.24.0)[70]. Centering of the normalized gene expression data was performed by subtracting the row means and scaling by dividing the columns by the standard deviation (SD) to generate a Z-score.

Differentially expressed genes between *STUB1*-deficient and wildtype cells were calculated with DESeq2[70] using FDR < 0.01.

**External datasets**. The anti-PD-1-treated melanoma patient samples were taken from Riaz et al.[12] (ENA/SRA database: PRJNA356761) and Gide et al.[71] (ENA/SRA database: PRJEB23709). The T cell-treated cell line data was taken from Vredevoogd et al.[19] (ENA/SRA database: SRP132830). Fastq files were downloaded and mapped to the human reference genome (Homo.sapiens.GRCh38.v82) using STAR(2.6.0c)[72] in 2-pass mode with default settings for paired-end data. The samples were used to generate read count data using HTSeq-count[73]. Normalization and statistical analysis of the expression of genes was performed using DESeq2[70]. Centering of the normalized gene expression data was performed by subtracting the row means and scaling by dividing the columns by the standard deviation (SD) to generate a Z-score. Clinical data were taken from the supplementary table from the original papers. Response to ICB was based on RECIST criteria as described in the papers (Responders: CR/PR/SD, Non-Responders: PD)[12,71]. To prevent confounding the correlation analysis by genes present in both gene sets, genes of the IFNγ receptor complex gene set (comprising IFNγ-R1, IFNγ-R2, JAK1, JAK2 and STAT1) that were present in the Hallmark IFNγ response gene set were removed from the Hallmark IFNγ response gene set prior to correlation analysis for the cell line analysis.

**Single cell data analysis**. Single cell RNAseq data on melanomas[38] was downloaded from the Single Cell Portal (accessed 20/05/2021), in which the reads were already normalized by TPM (GSE115978). Both the single cell RNAseq data sets on NSCLC (NSCLC_EMTAB6149)[37] and on BCC (BCC_GSE123813_aPD1)[39] were downloaded from the TISCH portal[74], in which the reads were already normalized and log-transformed. All three single cell data sets were loaded into Seurat (v4.0.2)[75] in R. We selected for the malignant cells in each single cell data set based on the already available annotation data. The signatures were calculated by taking the average of the genes in the signature for each cell. To prevent confounding the correlation analysis by genes present in both gene sets, genes of the IFNγ receptor complex gene set (comprising IFNγ-R1, IFNγ-R2, JAK1, JAK2 and STAT1) that were present in the Hallmark IFNγ response gene set were removed from the Hallmark IFNγ response gene set prior to correlation analysis.

**GSEA**. GSEAPreranked was performed using the BROAD javaGSEA standalone version (http://www.broadinstitute.org/gsea/downloads.jsp). Gene ranking was performed using the log$_2$-fold change in gene expression between D10 and SK-MEL-147 melanoma cells expressing either sgCtrl or sg*STUB1* that were treated with MART-1 T cells for 8 h (Supplementary Data 3). The pre-ranked gene list was run with 1000 permutations against the C2 canonical pathways. The full results of the GSEA are provided in Supplementary Data 4.

**Statistical analyses**. Statistical analyses for each experiment are indicated in the respective figure legends.

**Data collection and analysis software**. FACSDiva (v8.0), Graphpad Prism (v8.0.0 131), FlowJo (v10.6.0), GSEA (v4.0.3), MAGeCK (v0.5.7), R Studio (v1.1.463) with packages as indicated, Tophat v2.1, Perseus software (v1.5.6.0), Proteome Discoverer 2.2, STAR (2.6.0c). The codes used in this study are HTSeq-count[73], DESeq2[70] and Seurat (v4.0.2)[75].

**Reporting summary**. Further information on research design is available in the Nature Research Reporting Summary linked to this article.

## Data availability
The sequencing data of the genome-wide CRISPR/Cas9 screens in D10 and SK-MEL-23 melanoma cells is provided as Supplementary data 1, 2. The mass spectrometry data

generated in this study have been deposited in the Proteome Exchange database under accession code PXD030580. The RNA sequencing data have been deposited to the Gene Expression Omnibus under accession code GSE154040.

The anti-PD-1-treated melanoma patient samples were taken from Riaz et al.[12] (ENA/SRA database: PRJNA356761) and Gide et al.[71] (ENA/SRA database: PRJEB23709). The T cell-treated cell line data was taken from Vredevoogd et al.[19] (ENA/SRA database: SRP132830) [https://www.ebi.ac.uk/ena/browser/view/PRJNA434047?show=reads]. Single cell RNAseq data on melanomas[38] was downloaded from the Single Cell Portal (https://singlecell.broadinstitute.org/single_cell/study/SCP109/melanoma-immunotherapy-resistance#study-summary) (accessed 20/05/2021), in which the reads were already normalized by TPM (GEO: GSE115978). Both the single cell RNAseq data sets on NSCLC (NSCLC_EMTAB6149)[37], [https://www.ebi.ac.uk/arrayexpress/experiments/E-MTAB-6653/] and on BCC (BCC_GSE123813_aPD1)[39] [https://www.ncbi.nlm.nih.gov/geo/query/acc.cgi?acc=GSE123814] were downloaded from the TISCH portal[74] (http://tisch.comp-genomics.org/home/).

The remaining data are available within the article, supplementary information. Source data are provided with this paper.

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

## Acknowledgements

We thank all members of the Peeper lab as well as of the Division of Molecular Oncology and Immunology for constructive feedback and valuable input. We thank R. Mezzadra, C. Sun, M. Toebes, T. Schumacher, J. Staring, T. Brummelkamp, A. Thouin and J. Jacobs for sharing reagents and cell lines. Furthermore, we thank the flow cytometry, proteomics and sequencing core facilities as well as the animal housing facility and the animal pathology department of The Netherlands Cancer Institute for their support. O.B.B and M.A. acknowledge support of the X-omics Initiative, part of the NWO National Road-map for Large-Scale Research Infrastructures. J.D.L. was recipient of American Heart Grant 17POST33410945. J.C.S was recipient of National Institute on Aging Grant R01AG066710 and R01AG061188. D.S.P. is funded by the Oncode Institute, which is partly financed by the Dutch Cancer Society.

## Author contributions

G.A., D.W.V. and D.S.P. conceptualized the project, G.A. and D.W.V. performed the experiments and contributed equally to this work. O.B.B. and M.A. performed the proteomic profiling experiments. O.K., J.J.H.T., A.v.V. performed bioinformatic analyses for transcriptomic profiling. O.K. and O.B.B. contributed equally to this work. M.A.L., B.B., J.B. and C.P.L. carried out mouse experiments. D.D.A. and N.L.V. performed experiments. J.D.L. provided wildtype and mutant *IFNGR1*-ORF constructs. R.S.H., L.E.O., S.A. and J.C.S. provided tissues of CHIP-deficient mice. G.A., D.W.V. and D.S.P. wrote the manuscript. D.S.P. supervised this study. All authors have revised and approved the manuscript.

## Competing interests

D.S.P. is co-founder, shareholder and advisor of Immagene B.V.; M.A.L. is co-founder, shareholder and C.E.O. of Immagene B.V., which is unrelated to this study. O.K. is currently employed at Neogene Therapeutics B.V., which is unrelated to this study. The other authors declare no competing interests.
