## [Peer Review File · Nature Communications]

Reviewers' Comments:

Reviewer #1:

Remarks to the Author:

In this manuscript, authors tried to identify immune checkpoint pathways that stifle IFN- γ response. For this they hypothesize that increasing IFN- γ R1 would stimulate IFN- γ mediated cytotoxicity. Employing CRISPR/Cas9 screening system they observed that STUB1 could function as suppressor of IFN- γ R1 as well as JAK1. Moreover, they also observed that JAK1 could stabilize IFN- γ R1. Upon inactivation of STUB1, authors observed amplified IFN- γ signaling, stimulating sensitivity against cytotoxic T cells. Corroborating these observations there was increased of PD-L1. These correlations were further confirmed by STUB1 KO gene signature. They also provided a rationale for using PD-1 blockade system with STUB1 suppression employing mouse model which could probably be applicable for clinical application. The biochemical analyses of STUB1 E3 ligases targeting IFN- γ R1 and JAK1 are very limited and not comprehensive. Authors did not provide a direct interaction between STUB1 and its target proteins. More controls and experiments are required to support authors' hypothesis. The issues raised here are indicated below in more detail. In Figure 2d, e, h, and other immunoblot of IFN- γ R1 throughout the figures should be clearly labeled as authors mentioned the existence of glycosylated and basal forms. In figure 2e and f, the protein levels of IFN- γ R1 shown in e do not match with the expression levels indicated in figures. In addition, protein levels of IFN- γ R1 seem not to be increased in cells ectopically expressing IFN- γ R1 compared to others. In figure 2e, does the expression JAK1 stabilize the levels of STUB1? In figure 2f and g, the labels of Y axis are the same. Although description for each graph is mentioned in legend section, they should be more precisely indicated in several figures. The legend of figure 2i is wrongly explained.

In supplement figure 2d, the control cell with STUB1 is missing. Authors need to show that glycosylated IFN- γ R1 is actually increased upon STUB1 KO.

In figure 3, authors need to show domain IP interaction between STUB1 and IFN- γ R1 or JAK1. TPR deletion mutant of STUB1 not having any effects on its target could be due to various reasons including lack of protein-protein interaction, chaperone effects, etc. Which domains of STUB1 are required for the interactions?

Authors need to use STUB1 E3 ligase defective mutant, for example, H260Q mutant, to observe whether ubiquitination is really necessary step in mediating degradation of its targets. Also they could employ STUB1 point mutant not interacting with chaperones to study the effects of chaperones in these processes.

In the manuscript authors seem to argue that STUB1 recognizes ubiquitination of K285 on IFN- γ R1 and K249 on JAK1 and carry them to the proteasome. They also showed that these did not happen with TPR deletion mutant of STUB1. Furthermore, they showed that ubiquitination of IFN- γ R1 and JAK1 increased upon deletion of STUB1 by proteomics analyses, which seems to lead to the conclusion that STUB1 might not a direct E3 ligase of IFN- γ R1 and JAK2. For these arguments more comprehensive experiments are required.

- Authors need to show whether STUB1 and STUB1 TPR domain deletion mutant could bind to K285 on IFN- γ R1 and K249 on JAK1.

- They need to show whether the ubiquitination of IFN- γ R1 and JAK1 are K63 or K48 linked possibly using specific antibodies. The increase of ubiquitination of IFN- γ R1 and JAK1 could be K63 linked, which mainly involved in signaling pathways. It is very interesting to see whether the interaction of IFN- γ R1 and JAK1 could be increased upon STUB1 depletion. Also could the mutants of IFN- γ R1 or JAK1 affect the interaction between IFN- γ R1 and JAK1? Are the mutant forms of IFN- γ R1 and JAK1 more sensitive to IFN- γ -dependent cell toxicity? The question here is whether the ubiquitination process of IFN- γ R1 and JAK1 upon STUB1 deletion could be involved in cellular signaling leading to cytotoxicity.

- Ubiquitination process needs to be shown employing Ub analyses in addition to proteomics. The followings are recommended.

Does STUB1-dependent degradation of IFN- γ R1 or JAK1 ubiquitination-proteasome dependent pathways? The simple delivery of target proteins by STUB1 to proteasome complexes seems to be unknown so far and this needed to be proven. Authors need to show ubiquitination assay data as

follows:

- Ubiquitination using recombinant proteins and ubiquitin. If the targets are insoluble, they at least try to employ recombinant STUB1 and STUB1 H260Q enzyme activity defective mutant.
- Endogenous ubiquitination of IFN- γ R1 and JAK1 with or without STUB1.
- Reconstitution of STUB1 H260Q should be also carried out to prove that STUB1 could be directly involved in the ubiquitination process.
- These should be carried out in denaturation conditions.

In figure 3a and 3g it seems the levels of IFN- γ R1 and JAK1 seemed to be not increased upon MG132 treatment.

Figure 3b, fold changes should be presented in separate figures or normalized again as the bars in the graph are very misleading. It applies to other figures presented in a similar way.

In figure 3d and supple figure 3d, why are there no changes in the protein levels of IFN- γ R1 and JAK1 under the overexpression of STUB1. STUB1 should be also detected in this figure. In supple figure 3f, first panel, why is there less amount of K249R JAK1 in lane 5 compared to lane 6? In panel 2, there seems to be no change in the levels of JAK1 upon STUB1 KO.

In animal model figure 5e, the population of STUB1 deficient cells is very small compared to the control in NSG and immune-proficient mouse without PD-1 antibodies. Why does this happen in these mice? Does this mean that STUB1 KO leads to growth retardation of cancer cells? If so, then the other effects could play role in cells' sensitivity to cytotoxicity. Could reconstitution of STUB1 suppress cell death in vivo?

Reviewer #2:

Remarks to the Author:

Apriamashvili et al. followed the hypothesis that amplification of IFNG signaling in human melanoma cells could improve the outcome of T cell-based immunotherapy. In line with this, the authors found IFNGR1-high melanoma cells more susceptible to CD8+ T cells and IFNG compared to IFNGR1-low cells. Seeking for negative regulators of IFNGR1 they performed a genome-wide CRISPR/Cas9 screen and identified STUB1, an E3 ubiquitin ligase, targeting IFNGR1 and the associated kinase JAK1 for proteasomal degradation. Accordingly, inactivation of STUB1 increased IFNGR1/JAK1 levels in melanoma cells, which in turn amplified IFNG signaling and enhanced T cell sensitivity in vitro. Similar results were obtained with murine STUB1 (sgSTUB1) knock out melanoma cells (B16). B16-sgSTUB1 tumors grown in immunocompetent mice showed elevated PD-L1 expression levels compared to B16-sgCtrl tumors. Based on this observation, the authors performed in vivo competition assays, transplanting a mixture of B16-sgSTUB1 and B16-sgCtrl cells (1:1) onto immunodeficient and immunocompetent mice. The latter were treated or not with anti-PD1 antibodies. Tumors were harvested on day 12 and composition analyses revealed a clear decrease in the frequency of B16-sgSTUB1 cells in immunocompetent mice after anti-PD1 immunotherapy. Overall, the authors conclude that STUB1 inactivation (by pharmacological inhibitors) and anti-PD1 treatment should be combined to improve immunotherapy outcome.

Reviewer: The authors provide important new insights into the regulation of IFNGR1/JAK1 by the E3 ubiquitin ligase STUB1 and demonstrate that STUB1 inactivation sensitizes tumor cells to CD8+ T cells in vitro and in vivo which could be of therapeutic interest. Overall, the data are very convincing but some concerns remain.

Fig. 1d, e. Authors: „... We repeated this experiment with cytotoxic T cells, employing the matched tumor HLA-A*02:01+/MART1+ and 1D3 TCR T cell system we previously developed.”

Reviewer: The authors should be more detailed/precise in the description of their matched tumor-T cell system. Which tumor cells are used, D10? What is the specificity of the 1D3 TCR? In the Materials and Methods section they write as follows: „...MART1 retrovirus was made using a producer cell line as described previously ...”. The authors talk about the virus encoding the MART1 specific TCR, and not about a virus encoding MART-1 cDNA, correct?

Fig. 1d, e. Authors: „... In this experiment also, IFN γ -R1High melanoma cells showed higher susceptibility to T cell killing than IFN γ -R1Low cells.“

Reviewer: IFNGR1-high and IFNGR1-low melanoma cells should be controlled for MART1 and HLA-A*02:01 levels. It cannot be excluded that the sorted subpopulations show heterogenous expression which could impact on T cell killing.

Supplementary Fig. 1a , b. Authors: „... in a competition experiment, IFN γ treatment was two-fold more toxic to IFN γ -R1High than to IFN γ -R1Low cells.

Reviewer: data on IFNG-induced tumor cell death should be provided. Results from the competition assay could also reflect differences in cell proliferation under IFNG. Is cell cycle arrest induced by IFNG in both subpopulations? This comment addresses also data presented in Fig. 4h, i.

Based on the in vivo competition assay (Fig. 5e, f), the authors propose STUB1 as a therapeutic target.

Reviewer: The authors should analyze tumor growth and survival of mice challenged with either B16-sgSTUB1 or B16-sgCtrl cells and being treated with anti-PD1. Improved tumor control and survival upon STUB1 inactivation could strengthen its role as therapeutic target.

Fig. 4f. Authors: „... Differential gene expression of D10 and SK-MEL-147 melanoma cells lines after treatment with IFN γ for eight hours was used to derive an IFN γ response gene set ...“

Reviewer: The authors should include the list of IFN γ response genes as supplementary data.

Reviewer: Is the STUB1 expression level in pre-treatment melanoma biopsies associated with response to anti-PD1 therapy?

Introduction/Discussion:

Evolution of JAK1 deficiency in longitudinal clinical samples has been demonstrated by Sucker et al., Nat Commun 2017 (PMID: 28561041), showing data also on IFNG-induced cell death. This should be included in the introduction.

Several studies described a role for STUB1 in the regulation of IRF1, suggesting STUB1 could affect IFNG signaling also downstream of IFNGR1/JAK1. This should be discussed (PMID: 20947504).

Prolonged IFNG signaling could lead to resistance to immune checkpoint blockade as described by Benci et al., Cell 2016 (PMID: 27912061). The authors should discuss this in terms of the treatment schedule (anti-PD1 + STUB1 inhibitor).

Reviewer #3:

Remarks to the Author:

In this study Apriamashvili & Vredevoogd et al show that elevated expression of IFNgR will increase the sensitivity of tumor cells to IFNg and to T cells in a coculture system. The authors then perform a CRISPR screen to identify factors that regulate the cell surface expression of IFNgR and identify STUB1 and UROD 2 cell lines. They validate that both of these common hits, plus a cell line specific hit, AUP1, regulate the cell surface expression of IFNgR. The authors decide to follow up on STUB1 mechanistically. They validate that STUB1 increases IFNgR cell surface expression in a large number of cell lines and then demonstrate that STUB1 deletion increases both IFNgR and JAK1 protein but not transcript using proteomics and qPCR. They demonstrate that the effect of IFNgR destabilization is partially mediated by the destabilization of JAK1, which normally stabilizes IFNgR. They show that the degradation of both proteins is mediated ultimately by the proteasome and identify ubiquitin-modified lysine residues that STUB1 acts on to facilitate degradation of IFNgR and JAK1. The authors then go on to show that deletion of STUB1 greatly increases the sensitivity of tumor cells to IFN and T cells and that STUB1 deficient tumor cells are

depleted from competitive mixes in immunocompetent animals, especially when the PD-1 pathway is blocked. The authors end the paper by showing an association of the expression of STUB1-regulated genes and response to PD-1 blockade in melanoma.

-Major comments:

This body of work represents a significant advance for both our understanding of the regulation of interferon sensing and for cancer immunotherapy. Apriamashvili & Vredevogd et al show that STUB1 is a rational target for combination immunotherapy with PD-1 blockade and perform excellent cell biology and genetics experiments to elucidate the mechanism by which STUB1 regulates IFN sensing. The CRISPR screen that motivates the work on STUB1 was well-conceived and well-executed. The data in the manuscript is high quality. The manuscript is well-written and clear, the data is clearly presented and logically organized. Specific comments below:

1. One of the major messages of the paper is that STUB1 inhibition would be a rational combination strategy with PD-1 blockade, but the in vivo genetic evidence for this claim is shown only with a competition assay. The relative depletion of STUB1 KO cells in a mixture is strong evidence but is not exactly the same thing as demonstrating that an entirely STUB1-deficient tumor would respond more strongly to immunotherapy. The authors have all the reagents required to perform this in vivo study and clearly have the expertise to perform such in vivo studies. The absence of the experiment, given the claim in the abstract (Line 43-44), is a weakness of the paper. An experiment comparing the response of entirely STUB1 KO tumors to control tumors for tumor growth over time should be performed and the results should be reported, even if they are ambiguous. If STUB1 deletion does not make tumors overall respond better to PD-1 blockade, there could be additional mechanisms and the field should be aware of that.

2. Figure 5g-h: This data adds nothing to the manuscript and should probably just be removed. The STUB1 signature is very clearly a transcriptional signature of IFN response. Nearly half of the genes in the set are part of the Hallmark IFN gamma response signature, and the authors claim that any associations were not biased by the "limited presence of classical IFNg response genes in the STUB1-KO signature." I do not agree with this statement on a few different levels. First, it is a stretch to say that nearly half the genes coming from the Hallmark set represents a "limited presence." Second, the Hallmark IFNg gene set does not represent every IFN induced gene. I would bet that nearly all the genes in the STUB1 signature that are not in the Hallmark set are also unregulated in IFNg in control tumor cells, not just STUB1 KO tumor cells. The mechanism that the authors show for how STUB1 is working to enhance IFNg would be consistent with this result. If this were the case, then the authors would be showing that a set of IFN induced genes predicts response PD-1 blockade. This result has been demonstrated previously (PMID: 30388455; PMID: 28650338; PMID: 30309915; PMID: 25970248), and now Merck Pharmaceuticals uses a T cell inflamed gene expression profile, which overlaps heavily with an IFN response signature, as a biomarker for response to PD-1 inhibitors. This dataset seems to be a reflexive attempt to demonstrate human relevance, but the human cancer relevance has already been made clear previously. It is well known that IFN response signatures correlate with response to PD-1 blockade and the authors have identified a target that convincingly enhances the response to IFNg in tumors. The sort of patient cohort analysis shown in 5g-h is circular and not strengthening the manuscript. If the authors would like to include data showing this effect, a more serious effort must be made to understand if this is STUB1-specific or not. If it is not, that is completely fine, and it should either be removed or presented as additional evidence corroborating the importance of IFN signaling in tumors for PD-1 responses.

3. Figure 4: In the T cell/tumor cell co-cultures, is the depletion of STUB1 KO cells entirely because of sensitivity to IFNg or is there an additional effect from enhanced sensitivity to T cell killing, such as through upregulation of MHC-I?

-Minor comments:

1. The western blots in the figures are high quality, but it is always preferable to show densitometry quantification of the key results normalized to the loading control rather than leave the quantification up to the reader's eye. Please include densitometry for blots associated with key findings in the main figures.

2. Figure 2c - The proteomics analysis of STUB1 KO vs control SK-MEL-147 cells shows a protein that is increasing in STUB1 KO cells to a greater degree than IFN γ R and JAK1. What is this protein and why have the authors chosen not to show its identity or other top ranked differential proteins beyond just STUB1, IFN γ R and JAK1?

TO THE REVIEWERS:

Reviewer's Comments:

Reviewer #1:

In this manuscript, authors tried to identify immune checkpoint pathways that stifle IFN- γ response. For this they hypothesize that increasing IFN- γ R1 would stimulate IFN- γ -mediated cytotoxicity. Employing CRISPR/Cas9 screening system they observed that STUB1 could function as suppressor of IFN- γ R1 as well as JAK1. Moreover, they also observed that JAK1 could stabilize IFN- γ R1. Upon inactivation of STUB1, authors observed amplified IFN- γ signaling, stimulating sensitivity against cytotoxic T cells. Corroborating these observations there was increased of PD-L1. These correlations were further confirmed by STUB-1 KO gene signature. They also provided a rationale for using PD-1 blockade system with STUB1 suppression employing mouse model which could probably be applicable for clinical application.

The biochemical analyses of STUB1 E3 ligases targeting IFN- γ R1 and JAK1 are very limited and not comprehensive. Authors did not provide a direct interaction between STUB1 and its target proteins. More controls and experiments are required to support authors' hypothesis. The issues raised here are indicated below in more detail.

We thank the reviewer for their thoughtful and constructive comments and agree that additional biochemical assays would be valuable to further delineate the activity of STUB1. We thoroughly addressed the vast majority issues raised (in total, we included 33 new figure panels).

In Figure 2d, e, h, and other immunoblot of IFN- γ R1 throughout the figures should be clearly labeled as authors mentioned the existence of glycosylated and basal forms.

In the initial version of the manuscript, we had already indicated the different forms of IFN γ -R1 with three arrows next to the immunoblots themselves. However, to improve clarity, we now indicate this more clearly with a line next to the immunoblots and a description in the figure legends.

In figure 2e and f, the protein levels of IFN- γ R1 shown in e do not match with the expression levels indicated in figures.

The former Figure 2f did not represent the quantification of former Figure 2e, as the latter showed protein expression of JAK1, IFN γ -R1 and STUB1 on an immunoblot. The former panel 2f showed the quantification of IFN γ -R1 expression using flow cytometry. We have now clarified the former panels Figure 2e-g by indicating the different readouts (new Figure 2c-e).

In addition, protein levels of IFN- γ R1 seem not to be increased in cells ectopically expressing IFN- γ R1 compared to others.

Importantly, RNA levels of *IFNGR1* are strongly increased upon ectopic expression (new Figure 2e). However, as the reviewer points out, this does not translate into increased protein levels of IFN γ -R1. This observation is indicative of the strong dependency of IFN γ -R1 protein stability on the levels of JAK1 protein as we describe in the manuscript (see also new Figure 2c-e).

In figure 2e, does the expression JAK1 stabilize the levels of STUB1?

JAK1 overexpression and/or genetic ablation does not affect STUB1 protein levels, as can be observed from the new Figure 2f and Reviewer Figure 1g.

In figure 2f and g, the labels of Y axis are the same. Although description for each graph is mentioned in legend section, they should be more precisely indicated in several figures.

The axis label in former Figure 2f (now Figure 2d) referred to protein expression of IFN γ -R1 while the label in former Figure 2g (now Figure 2e) referred to the mRNA expression of *IFNGR1*. To improve clarity, we now clearly stated the readout method above each panel (new Figure 2c-e).

The legend of figure 2i is wrongly explained.

We thank the reviewer for pointing out this mistake; it has now been corrected.

In supplement figure 2d, the control cell with STUB1 is missing. Authors need to show that glycosylated IFN- γ R1 is actually increased upon STUB1 KO.

We have now added the whole cell lysate samples used as input samples for all three conditions to the IP experiment in new Supplementary Figure 2d. In new Supplementary Figure 2c we show a quantification of the low and high molecular weight species of IFN γ -R1, demonstrating that the high molecular weight bands are more abundant in STUB1 KO cells.

In figure 3, authors need to show domain IP interaction between STUB1 and IFN- γ R1 or JAK1. TPR deletion mutant of STUB1 not having any effects on its target could be due to various reasons including lack of protein-protein interaction, chaperone effects, etc. Which domains of STUB1 are required for the interactions? Authors need to use STUB1 E3 ligase defective mutant, for example, H260Q mutant, to observe whether ubiquitination is really necessary step in mediating degradation of its targets. Also they could employ STUB1 point mutant not interacting with chaperones to study the effects of chaperones in these processes.

We thank the referee for these suggestions and have included the suggested experiment in new Figures 2i-k. The results show that STUB1 requires both its TPR domain as well as its E3 ligase function for the proper regulation of IFN γ -R1 and JAK1.

In the manuscript authors seem to argue that STUB1 recognizes ubiquitination of K285 on IFN- γ R1 and K249 on JAK1 and carry them to the proteasome. They also showed that these did not happen with TPR deletion mutant of STUB1. Furthermore, they showed that ubiquitination of IFN- γ R1 and JAK1 increased upon deletion of STUB1 by proteomics analyses, which seems to lead to the conclusion that STUB1 might not a direct E3 ligase of IFN- γ R1 and JAK2. For these arguments more comprehensive experiments are required. Authors need to show whether STUB1 and STUB1 TPR domain deletion mutant could bind to K285 on IFN- γ R1 and K249 on JAK1.

We have now performed the suggested experiment. In new Figure 2i we show that the interaction of STUB1 with IFN γ -R1 and JAK1 is dependent on the TPR domain. We also demonstrate that STUB1 requires E3 ubiquitin ligase activity for its regulation of IFN γ -R1 and JAK1 (new Figures 2j and k). Finally, we now show that JAK1 can be directly ubiquitinated by STUB1 on residue K249 (new Figure 3j). In the previous version we showed the direct comparison of ubiquitinated peptides in STUB1-KO cells relative to wildtype cells (former Figure 3e). This comparison however, did not take into consideration the increased protein levels of JAK1 (new Figure 2a, b), which confound this particular comparison. We have now corrected for JAK1 protein levels in the ubiproteome dataset (described in the Materials and Methods), resulting in a measure of ubiquitinated peptides relative to the total protein abundance, showing that JAK1^{K249} to be relatively less ubiquitinated in STUB1-depleted cells (new Figure 3k).

They need to show whether the ubiquitination of IFN- γ R1 and JAK1 are K63 or K48 linked possibly using specific antibodies. The increase of ubiquitination of IFN- γ R1 and JAK1 could be K63 linked, which mainly involved in signaling pathways.

We have now performed the suggested experiment. By using different ubiquitin mutants, we were able to delineate which type of ubiquitin linkage is required for STUB1-mediated regulation of JAK1. Comparison of JAK1 expression in STUB1-deficient and -proficient cells in the context of the different lysine mutants revealed that K63-mediated ubiquitination is sufficient for STUB1-mediated regulation of JAK1 (Reviewer Figure 1a). For IFN γ -R1, the effects were less clear, as neither linkage type was sufficient nor required (Reviewer Figure 1a). This may be due to a requirement of a combined, possibly branched, K48/K63 linkage¹⁻⁵, or to a different ubiquitin lysine linkage type requirement altogether. There may also be additional regulation of IFN γ -R1 by JAK1, as demonstrated in Figure 2.

It is very interesting to see whether the interaction of IFN- γ R1 and JAK1 could be increased upon STUB1 depletion.

The reviewer raises an interesting and important point. The vast majority of our experiments assessing the effects of *STUB1* inactivation on IFN γ -R1 and JAK1 stability were performed under baseline conditions without additional treatments applied. To what extent the IFN γ -R complex is already pre-assembled in the absence of IFN γ is not fully understood⁶. In an attempt to address this point, we have performed immunoprecipitation for both IFN γ -R1 and JAK1 (Reviewer Figure 1b), but could not detect the respective interaction partner at baseline. Thus, although we made a serious attempt, we were unable to conclusively address this question.

Also could the mutants of IFN- γ R1 or JAK1 affect the interaction between IFN- γ R1 and JAK1?

Although the reviewer raises an intriguing question, given our result described above, we cannot address this question conclusively, unfortunately.

*Are the mutant forms of IFN- γ R1 and JAK1 more sensitive to IFN- γ -dependent cell toxicity? The question here is whether the ubiquitination process of IFN- γ R1 and JAK1 upon *STUB1* deletion could be involved in cellular signaling leading to cytotoxicity.*

We now performed the suggested experiment. Melanoma cells expressing IFN γ -R1^{K285R} and JAK1^{K249R} show a mildly increased sensitivity to IFN γ treatment compared to cells expressing wildtype versions of these proteins (Reviewer Figure 1c).

*Does *STUB1*-dependent degradation of IFN- γ R1 or JAK1 ubiquitination-proteasome dependent pathways?*

We had already shown in the first version of the manuscript that IFN γ -R1 and JAK1 are proteasomally degraded by *STUB1* (now shown in new Figure 3a-c and Supplementary Figure 3a-c).

*The simple delivery of target proteins by *STUB1* to proteasome complexes seems to be unknown so far and this needed to be proven. Authors need to show ubiquitination assay data as follows: Ubiquitination using recombinant proteins and ubiquitin. If the targets are insoluble, they at least try to employ recombinant *STUB1* and *STUB1* H260Q enzyme activity defective mutant. Endogenous ubiquitination of IFN- γ R1 and JAK1 with or without *STUB1*.*

We have now included an *in vitro* ubiquitination assay in new Figure 3j as proposed by the reviewer, showing direct ubiquitination of JAK1^{K249} by *STUB1*. Additionally, we now demonstrate that *STUB1* requires both the TPR domain and the E3 ubiquitin ligase activity to regulate IFN γ -R1 and JAK1 (new Figures 2i-j). And importantly, we were now able to show direct ubiquitination of JAK1^{K249} by *STUB1* (new Figure 3j) and also that *STUB1*-deficient cells exhibit a relative reduction of JAK1^{K249} ubiquitination levels (new Figure 3k).

*Reconstitution of *STUB1* H260Q should be also carried out to prove that *STUB1* could be directly involved in the ubiquitination process.*

We thank the referee for these suggestions and have included the suggested experiment in new Figures 2i and j. The result shows that the E3 ligase-deficient *STUB1* variant is unable to regulate IFN γ -R1 and JAK1 protein levels (new Figure 3j and k).

These should be carried out in denaturation conditions.

In line with the reviewer's recommendations, we performed the *in vitro* ubiquitination assay under denaturing conditions (new Figure 3j).

In figure 3a and 3g it seems the levels of IFN- γ R1 and JAK1 seemed to be not increased upon MG132 treatment.

Quantifications of IFN γ -R1 and JAK1 protein levels for the indicated immunoblots were already present in the previous version of the manuscript. The indicated quantifications from new Figure 3a are shown

in new Figure 3b and c, respectively. The quantifications of IFN γ -R1 and JAK1 protein levels in new Figure 3e are shown in new Figure 3f and g, respectively.

Figure 3b, fold changes should be presented in separate figures or normalized again as the bars in the graph are very misleading. It applies to other figures presented in a similar way.

We have incorporated the referee's suggestions and represented each genotype condition as a separate graph (new Figure 3b, c, f, g, i).

In figure 3d and supple figure 3d, why are there no changes in the protein levels of IFN- γ R1 and JAK1 under the overexpression of STUB1.

We used different complementary approaches to assess how STUB1 regulates IFN γ -R1 and JAK1 expression, aiming to answer fundamentally different questions. On the one hand we genetically inactivated STUB1 to ask whether it is required for limiting IFN γ -R1/JAK1 levels, which is the case (e.g. new Figures 2a-b). Conversely, we have overexpressed STUB1 in STUB1-proficient cells to query whether STUB1 is sufficient to limit IFN γ -R1/JAK1 levels, which is not the case (Reviewer Figures 1d-f).

STUB1 should be also detected in this figure.

In line with the reviewer's recommendations, we have now included an immunoblot for STUB1, to supplement the detection of overexpressed STUB1 by a FLAG immunoblot. We have now also visualized STUB1 in the same blot (new Figure 2k). Of note, the STUB1 antibody we have used recognizes the N-terminal portion of STUB1, which is absent when removing the N-terminal TPR domain. For this reason, we can detect FLAG-tagged STUB1 lacking its TPR domain, while this band is absent when detecting STUB1 using the antibody for the native protein.

In supple figure 3f, first panel, why is there less amount of K249R JAK1 in lane 5 compared to lane 6?

STUB1 inactivation causes a strong increase in JAK1 protein levels (lane 3 vs. 4). The mutation of JAK1^{K249R} is also causes a significant increase in JAK1 protein stability (lane 3 vs. 5). Inactivation of STUB1 in the context of this mutation appears to still have a minor effect on protein stability. In panel 2, there seems to be no change in the levels of JAK1 upon STUB1 KO. It is important to keep in mind however, that the entire left panel of this blot lacks IFN γ -R1 expression. This is important because this appears to have a negative effect on JAK1 protein levels (compare lane 3 vs. 9). Similar to our observation the increased IFN γ -R1 protein levels in STUB1-KO cells is largely JAK1-dependent (new Figure 2f and g), this implies that increased JAK1 protein levels in STUB1-KO cells are in part also dependent on IFN γ -R1 (compare lane 4 vs.10). To account for this complex mode of regulation and interdependence, we chose to only compare the effects of STUB1-inactivation on the combination of either the wildtype IFN γ -R1/JAK1 or the respective lysine-arginine mutants in the main figure (see new Figure 3h, I and new Supplementary Figure 3e and f).

For the sake of full transparency, we provide the more complex blot containing all combinations (Reviewer Figure 1g). Given the complex nature of this blot we would propose to remove it from the manuscript to avoid confusion, yet we provide it to the referees for the proper assessment of the data (see Reviewer Figure 1g).

In animal model figure 5e, the population of STUB1 deficient cells is very small compared to the control in NSG and immune-proficient mouse without PD-1 antibodies. Why does this happen in these mice? Does this mean that STUB1 KO leads to growth retardation of cancer cells? If so, then the other effects could play role in cells' sensitivity to cytotoxicity. Could reconstitution of STUB1 suppress cell death in vivo?

The reviewer is correct in noting that STUB1 KO cells seem to grow slower than their wildtype counterparts in the B16F10-dOVA cell line *in vivo* in the absence of a functional immune system. This has also been observed by others⁷. We are unsure why this happens. However, by normalizing to this condition in the mixing experiment, we were able to analyze specifically the immune-induced sensitivity and thereby circumvent these growth defects.

Lastly, we have updated the manuscript title and summary to better reflect the conclusions of the revised manuscript, with less emphasis on the clinical relevance and more on the mechanistic insight and the complexity of STUB1-regulated IFN γ signaling.

Reviewer #2:

Apriamashvili et al. followed the hypothesis that amplification of IFNG signaling in human melanoma cells could improve the outcome of T cell-based immunotherapy. In line with this, the authors found IFNGR1-high melanoma cells more susceptible to CD8+ T cells and IFNG compared to IFNGR1-low cells. Seeking for negative regulators of IFNGR1 they performed a genome-wide CRISPR/Cas9 screen and identified STUB1, an E3 ubiquitin ligase, targeting IFNGR1 and the associated kinase JAK1 for proteasomal degradation. Accordingly, inactivation of STUB1 increased IFNGR1/JAK1 levels in melanoma cells, which in turn amplified IFNG signaling and enhanced T cell sensitivity in vitro. Similar results were obtained with murine STUB1 (sgSTUB1) knock out melanoma cells (B16). B16-sgSTUB1 tumors grown in immunocompetent mice showed elevated PD-L1 expression levels compared to B16-sgCtrl tumors. Based on this observation, the authors performed in vivo competition assays, transplanting a mixture of B16-sgSTUB1 and B16-sgCtrl cells (1:1) onto immunodeficient and immunocompetent mice. The latter were treated or not with anti-PD1 antibodies. Tumors were harvested on day 12 and composition analyses revealed a clear decrease in the frequency of B16-sgSTUB1 cells in immunocompetent mice after anti-PD1 immunotherapy. Overall, the authors conclude that STUB1 inactivation (by pharmacological inhibitors) and anti-PD1 treatment should be combined to improve immunotherapy outcome.

The authors provide important new insights into the regulation of IFNGR1/JAK1 by the E3 ubiquitin ligase STUB1 and demonstrate that STUB1 inactivation sensitizes tumor cells to CD8+ T cells in vitro and in vivo which could be of therapeutic interest. Overall, the data are very convincing but some concerns remain.

We thank this reviewer for their enthusiasm and helpful and constructive comments.

Fig. 1d, e. Authors: „... We repeated this experiment with cytotoxic T cells, employing the matched tumor HLA-A*02:01+/MART1+ and 1D3 TCR T cell system we previously developed.“
Reviewer: The authors should be more detailed/precise in the description of their matched tumor-T cell system. Which tumor cells are used, D10? What is the specificity of the 1D3 TCR? In the Materials and Methods section they write as follows: „...MART1 retrovirus was made using a producer cell line as described previously ...“. The authors talk about the virus encoding the MART1 specific TCR, and not about a virus encoding MART-1 cDNA, correct?

We have incorporated the feedback of referee #2 and have clarified in the text accordingly.

Fig. 1d, e. Authors: „... In this experiment also, IFN γ -R1^{High} melanoma cells showed higher susceptibility to T cell killing than IFN γ -R1^{Low} cells.“
Reviewer: IFNGR1-high and IFNGR1-low melanoma cells should be controlled for MART1 and HLA-A*02:01 levels. It cannot be excluded that the sorted subpopulations show heterogenous expression which could impact on T cell killing.

In line with the reviewer's recommendations, we have now included experiments addressing this potential confounder: IFN γ -R1^{high} and IFN γ -R1^{low}-expressing cells have similar levels of MHC class I expression prior to treatment (new Supplementary Figure 1b and Reviewer Figure 1 h). Additionally, blocking IFN γ in a co-culture with T cells and either IFN γ -R1^{high} or IFN γ -R1^{low}-expressing cells almost entirely rescues the sensitization effect we observed. This experiment indicates that IFN γ -R1^{high}-expressing cells are sensitized to cytotoxic T cells primarily through enhanced IFN γ signaling (new Figure 1f and Reviewer Figure 1h).

Supplementary Fig. 1a, b. Authors: „... in a competition experiment, IFN γ treatment was two-fold more toxic to IFN γ -R1^{High} than to IFN γ -R1^{Low} cells.

Reviewer: data on IFNG-induced tumor cell death should be provided. Results from the competition assay could also reflect differences in cell proliferation under IFNG. Is cell cycle arrest induced by IFNG in both subpopulations? This comment addresses also data presented in Fig. 4h, i.

In response to this comment, we have done several proliferation and cell death experiments but could not single out an effect that stood out. This is consistent with the large number of negative biological effects on tumors reported for IFN γ , ranging from ER stress, cell cycle arrest to cell death⁸⁻¹¹. We have changed our wording regarding in the text accordingly.

Based on the in vivo competition assay (Fig. 5e, f), the authors propose STUB1 as a therapeutic target. Reviewer: The authors should analyze tumor growth and survival of mice challenged with either B16-sgSTUB1 or B16-sgCtrl cells and being treated with anti-PD1. Improved tumor control and survival upon STUB1 inactivation could strengthen its role as therapeutic target.

We thank this reviewer for their enthusiasm and helpful and constructive comments.

To address the comment of this (and the other) referee for more information on the potential for translation of our *in vivo* data in the context of anti-PD-1 therapy, we have performed the requested *in vivo* experiment. It was carried out in the same syngeneic B16F10-dOVA murine melanoma model. We transplanted either sgCtrl- or sgStub1-expressing melanoma cells into syngeneic mice and initiated anti-PD-1 treatment once tumors reach an average size of 100 mm³ (New Supplementary Fig. 5i). As opposed to the competition experiment, we did not observe an enhanced response to anti-PD-1 treatment. The results of the competition experiment recapitulate previously reported results by Manguso *et al.* on Stub1⁷, and extend the observation that IFN γ -insensitive tumors are sensitive to immune pressure unless protected by PD-L1 on bystander cells and only gain an advantage in the context of PD-L1-targeting immunotherapy¹². In our mixing model we observe the converse phenomenon, namely that enhanced IFN γ signaling becomes detrimental only in the context of anti-PD-1 treatment. The results from the new *in vivo* experiment in which full B16F10-dOVA Stub1 knockout tumors failed to show enhanced anti-PD1 response extends the observations made by Williams *et al.*¹². It could also be indicative of the phenomenon that prolonged IFN γ signaling can give rise to PD-L1-independent resistance mechanisms¹³, as well as creating a more immune-suppressive TME¹⁴. This point was also raised by reviewer #2 and is now also discussed in the manuscript. We now clearly state that for any translation of STUB1 inhibition, it will be important to deepen our understanding of the effects of IFN γ signaling *in vivo* to better predict in which context this combination would be most beneficial¹⁵. Therefore, we agree with the assessment of reviewer #3 and would like to report these data to merit future investigations on this topic. We have included a section to carefully describe this in the discussion.

Fig. 4f. Authors: "... Differential gene expression of D10 and SK-MEL-147 melanoma cells lines after treatment with IFN γ for eight hours was used to derive an IFN γ response gene set ..."
Reviewer: The authors should include the list of IFN γ response genes as supplementary data.

The panel the reviewer is referring to was deemed redundant in view of the gene set enrichment analysis (GSEA) and was removed. We have now extended our GSEA analysis, by performing a broader GSE analysis on the T cell response of STUB1-deficient cells and have included the most strongly enriched gene sets (FDR<0.05) in new Figure 4d, e and new Supplementary Figure 4e. Additionally we provided a Supplementary table containing the complete list of pathways of the GSEA analysis.

Reviewer: Is the STUB1 expression level in pre-treatment melanoma biopsies associated with response to anti-PD1 therapy?

We now performed the suggested experiment. STUB1 mRNA expression prior to immunotherapy is not associated with better response to anti-PD-1 (Reviewer Fig. 1i).

Evolution of JAK1 deficiency in longitudinal clinical samples has been demonstrated by Sucker et al., Nat Commun 2017 (PMID: 28561041), showing data also on IFNG-induced cell death. This should be included in the introduction. Several studies described a role for STUB1 in the regulation of IRF1, suggesting STUB1 could affect IFNG signaling also downstream of IFNGR1/JAK1. This should be discussed (PMID: 20947504). Prolonged IFNG signaling could lead to resistance to immune checkpoint blockade as described by Benci et al., Cell 2016 (PMID: 27912061). The authors should discuss this in terms of the treatment schedule (anti-PD1 + STUB1 inhibitor).

We agree with the suggestions made by the referee and have incorporated these references in the introduction and discussion.

Lastly, we have updated the manuscript title and summary to better reflect the conclusions of the revised manuscript, with less emphasis on the clinical relevance and more on the mechanistic insight and the complexity of STUB1-regulated IFN γ signaling.

Reviewer #3:

One of the major messages of the paper is that STUB1 inhibition would be a rational combination strategy with PD-1 blockade, but the in vivo genetic evidence for this claim is shown only with a competition assay. The relative depletion of STUB1 KO cells in a mixture is strong evidence but is not exactly the same thing as demonstrating that an entirely STUB1-deficient tumor would respond more strongly to immunotherapy. The authors have all the reagents required to perform this in vivo study and clearly have the expertise to perform such in vivo studies. The absence of the experiment, given the claim in the abstract (Line 43-44), is a weakness of the paper. An experiment comparing the response of entirely STUB1 KO tumors to control tumors for tumor growth over time should be performed and the results should be reported, even if they are ambiguous. If STUB1 deletion does not make tumors overall respond better to PD-1 blockade, there could be additional mechanisms and the field should be aware of that.

We thank this reviewer for their enthusiasm and helpful and constructive comments.

To address the comment of this (and the other) referee for more information on the potential for translation of our *in vivo* data in the context of anti-PD-1 therapy, we have performed the requested *in vivo* experiment. It was carried out in the same syngeneic B16F10-dOVA murine melanoma model. We transplanted either sgCtrl- or sgStub1-expressing melanoma cells into syngeneic mice and initiated anti-PD-1 treatment once tumors reach an average size of 100 mm³ (New Supplementary Fig. 5i). As opposed to the competition experiment, we did not observe an enhanced response to anti-PD-1 treatment. The results of the competition experiment recapitulate previously reported results by Manguso *et al.* on *Stub1*⁷, and extend the observation that IFN γ -insensitive tumors are sensitive to immune pressure unless protected by PD-L1 on bystander cells and only gain an advantage in the context of PD-L1-targeting immunotherapy¹². In our mixing model we observe the converse phenomenon, namely that enhanced IFN γ signaling becomes detrimental only in the context of anti-PD-1 treatment. The results from the new *in vivo* experiment in which full B16F10-dOVA *Stub1* knockout tumors failed to show enhanced anti-PD1 response extends the observations made by Williams *et al.*¹². It could also be indicative of the phenomenon that prolonged IFN γ signaling can give rise to PD-L1-independent resistance mechanisms¹³, as well as creating a more immune-suppressive TME¹⁴. This point was also raised by reviewer #2 and is now also discussed in the manuscript. We now clearly state that for any translation of STUB1 inhibition, it will be important to deepen our understanding of the effects of IFN γ signaling *in vivo* to better predict in which context this combination would be most beneficial¹⁵. Therefore, we agree with the assessment of reviewer #3 and would like to report these data to merit future investigations on this topic. We have included a section to carefully describe this in the discussion.

Figure 5g-h: This data adds nothing to the manuscript and should probably just be removed. The STUB1 signature is very clearly a transcriptional signature of IFN response. Nearly half of the genes in the set are part of the Hallmark IFN gamma response signature, and the authors claim that any associations were not biased by the "limited presence of classical IFN γ response genes in the STUB1-KO signature." I do not agree with this statement on a few different levels. First, it is a stretch to say that nearly half the genes coming from the Hallmark set represents a "limited presence." Second, the Hallmark IFN γ gene set does not represent every IFN induced gene. I would bet that nearly all the genes in the STUB1 signature that are not in the Hallmark set are also unregulated in IFN γ in control tumor cells, not just STUB1 KO tumor cells.

The mechanism that the authors show for how STUB1 is working to enhance IFN γ would be consistent with this result. If this were the case, then the authors would be showing that a set of IFN induced genes predicts response PD-1 blockade. This result has been demonstrated previously (PMID: 30388455; PMID: 28650338; PMID: 30309915; PMID: 25970248), and now Merck Pharmaceuticals uses a T cell inflamed gene expression profile, which overlaps heavily with an IFN response signature, as a

biomarker for response to PD-1 inhibitors. This dataset seems to be a reflexive attempt to demonstrate human relevance, but the human cancer relevance has already been made clear previously. It is well known that IFN response signatures correlate with response to PD-1 blockade and the authors have identified a target that convincingly enhances the response to IFN γ in tumors. The sort of patient cohort analysis shown in 5g-h is circular and not strengthening the manuscript. If the authors would like to include data showing this effect, a more serious effort must be made to understand if this is STUB1-specific or not. If it is not, that is completely fine, and it should either be removed or presented as additional evidence corroborating the importance of IFN signaling in tumors for PD-1 responses.

We partially agree with the comments of the reviewer regarding the enrichment of IFN γ response genes in the STUB1-KO signature and the drawbacks of this approach. Specifically, given the overlap between IFN γ signaling components and the STUB1-KO signature, we may not be able to truly dissect the two and assess the relative importance of either signature. We therefore removed the respective panels from the revised manuscript. The data we present in this study positions STUB1 as an important negative regulator of IFN γ signaling. To better address the significance of this mode of regulation in a relevant patient population, we correlated STUB1 expression with IFN γ response and found a striking negative correlation (new Figure 4l and new Supplementary Figure 4l).

Figure 4: In the T cell/tumor cell co-cultures, is the depletion of STUB1 KO cells entirely because of sensitivity to IFN γ or is there an additional effect from enhanced sensitivity to T cell killing, such as through upregulation of MHC-I?

The enhanced sensitivity of STUB1-KO tumors to MART-1 T cells is primarily due to the enhanced sensitivity to IFN γ , since the ablation of IFN γ signaling completely rescues this effect (new Figures 4j, k and new Supplementary Figures 4j and k, new Supplementary Figure 5d and e). While this shows that the enhanced sensitivity is IFN γ -dependent, we cannot exclude that the regulation of MHC-I by IFN γ may contribute. We have therefore also included a sentence in the manuscript to indicate this.

The western blots in the figures are high quality, but it is always preferable to show densitometry quantification of the key results normalized to the loading control rather than leave the quantification up to the reader's eye. Please include densitometry for blots associated with key findings in the main figures.

The quantifications of the immunoblots that were included in the first version of the manuscript had all been quantified using the densitometry method and normalized to the loading control already. We now more clearly indicate this both in the Figure legends and in the Methods.

Figure 2c - The proteomics analysis of STUB1 KO vs control SK-MEL-147 cells shows a protein that is increasing in STUB1 KO cells to a greater degree than IFN γ R and JAK1. What is this protein and why have the authors chosen not to show its identity or other top ranked differential proteins beyond just STUB1, IFN γ R and JAK1?

In this figure we only highlighted proteins that were up- or downregulated in both cell lines. We have now included a sentence in the manuscript to emphasize this. We have now also included a table with the mass spectrometry data to be fully transparent about other proteins readers may be interested in.

Lastly, we have updated the manuscript title and summary to better reflect the conclusions of the revised manuscript, with less emphasis on the clinical relevance and more on the mechanistic insight and the complexity of STUB1-regulated IFN γ signaling.

References

1. Meyer HJ, Rape M. Enhanced Protein Degradation by Branched Ubiquitin Chains. *Cell*. 2014;157(4):910-921. doi:10.1016/J.CELL.2014.03.037
2. Ohtake F, Saeki Y, Ishido S, Kanno J, Tanaka K. The K48-K63 Branched Ubiquitin Chain Regulates NF- κ B Signaling. *Mol Cell*. 2016;64(2):251-266. doi:10.1016/J.MOLCEL.2016.09.014
3. Yau RG, Doerner K, Castellanos ER, et al. Assembly and Function of Heterotypic Ubiquitin Chains in Cell-Cycle and Protein Quality Control. *Cell*. 2017;171(4):918-933.e20. doi:10.1016/J.CELL.2017.09.040
4. Liu C, Liu W, Ye Y, Li W. Ufd2p synthesizes branched ubiquitin chains to promote the degradation of substrates modified with atypical chains. *Nat Commun*. 2017;8. doi:10.1038/NCOMMS14274
5. Ohtake F, Tsuchiya H, Saeki Y, Tanaka K. K63 ubiquitylation triggers proteasomal degradation by seeding branched ubiquitin chains. *Proc Natl Acad Sci U S A*. 2018;115(7):E1401-E1408. doi:10.1073/PNAS.1716673115/-/DCSUPPLEMENTAL
6. Blouin CM, Lamaze C. Interferon gamma receptor: The beginning of the journey. *Front Immunol*. 2013;4(SEP):267. doi:10.3389/FIMMU.2013.00267/BIBTEX
7. Manguso RT, Pope HW, Zimmer MD, et al. In vivo CRISPR screening identifies Ptpn2 as a cancer immunotherapy target. *Nature*. 2017;547(7664):413-418. doi:10.1038/nature23270
8. Gooch JL, Herrera RE, Yee D. The role of p21 in interferon γ -mediated growth inhibition of human breast cancer cells. *Cell Growth Differ*. 2000;11(6):335-342.
9. Dai C, Krantz SB. Interferon γ induces upregulation and activation of caspases 1, 3, and 8 to produce apoptosis in human erythroid progenitor cells. *Blood*. 1999;93(10):3309-3316. doi:10.1182/blood.v93.10.3309.410k04_3309_3316
10. Siegmund D, Wicovsky A, Schmitz I, et al. Death Receptor-Induced Signaling Pathways Are Differentially Regulated by Gamma Interferon Upstream of Caspase 8 Processing. *Mol Cell Biol*. 2005;25(15):6363-6379. doi:10.1128/mcb.25.15.6363-6379.2005
11. Fulda S, Debatin KM. IFN γ sensitizes for apoptosis by upregulating caspase-8 expression through the Stat1 pathway. *Oncogene*. 2002;21(15):2295-2308. doi:10.1038/sj.onc.1205255
12. Williams JB, Li S, Higgs EF, et al. Tumor heterogeneity and clonal cooperation influence the immune selection of IFN- γ -signaling mutant cancer cells. *Nat Commun*. 2020;11(1):1-14. doi:10.1038/s41467-020-14290-4
13. Benci JL, Xu B, Qiu Y, et al. Tumor Interferon Signaling Regulates a Multigenic Resistance Program to Immune Checkpoint Blockade. *Cell*. 2016;167(6):1540-1554.e12. doi:10.1016/j.cell.2016.11.022
14. Benci JL, Johnson LR, Choa R, et al. Opposing Functions of Interferon Coordinate Adaptive and Innate Immune Responses to Cancer Immune Checkpoint Blockade. *Cell*. 2019;178(4):933-948.e14. doi:10.1016/j.cell.2019.07.019
15. Vredevogd DW, Apriamashvili G, Peeper DS. The (re)discovery of tumor-intrinsic determinants of immune sensitivity by functional genetic screens. *Immuno-Oncology Technol*. 2021;11:100043. doi:10.1016/J.IOTECH.2021.100043

Reviewers' Comments:

Reviewer #1:

Remarks to the Author:

Authors addressed the issues raised in the revised manuscript.

There is no further comment.

Reviewer #2:

Remarks to the Author:

The authors addressed all points that I raised, including in vivo studies. The new in vivo results highlight the context-dependent role of tumor cell-intrinsic STUB1 in anti-PD-1 therapy.

Reviewer #3:

Remarks to the Author:

The authors have strengthened the manuscript with the addition of more experimental data supporting their claims on the mechanism of STUB1 mediated degradation of IFNGR1 and JAK1. In addition the authors removed an analysis of expression signatures from patient data that was potentially confounded, and performed an in vivo validation study using full STUB1 KO tumors. The title and abstract have been modified to reflect the current focus of the narrative. I applaud the authors for their rigorous approach to the revision of the manuscript and I believe that it is improved significantly. All of my questions and concerns have been addressed and I fully support publication.

TO THE REVIEWERS:

Reviewer #1 (Remarks to the Author):

Authors addressed the issues raised in the revised manuscript. There is no further comment.

We thank the reviewer for his/her valuable suggestions that helped to improve the mechanistic aspects of our manuscript considerably.

Reviewer #2 (Remarks to the Author):

The authors addressed all points that I raised, including in vivo studies. The new in vivo results highlight the context-dependent role of tumor cell-intrinsic STUB1 in anti-PD-1 therapy.

We thank the reviewer for his/her constructive comments to clarify the translational insights of our manuscript.

Reviewer #3 (Remarks to the Author):

The authors have strengthened the manuscript with the addition of more experimental data supporting their claims on the mechanism of STUB1 mediated degradation of IFNGR1 and JAK1. In addition the authors removed an analysis of expression signatures from patient data that was potentially confounded, and performed an in vivo validation study using full STUB1 KO tumors. The title and abstract have been modified to reflect the current focus of the narrative. I applaud the authors for their rigorous approach to the revision of the manuscript and I believe that it is improved significantly. All of my questions and concerns have been addressed and I fully support publication.

We thank the reviewer for his/her helpful and constructive comments that helped us to improve the clinical analyses presented as well as the translational implications of the study.